# DELAYED LOCAL-SGD FOR DISTRIBUTED LEARNING WITH LINEAR SPEEDUP

## ABSTRACT

*Local-SGD*-based algorithms have gained much popularity in distributed learning to reduce the communication overhead, where each client conducts multiple localized iterations before communicating with the central server. However, since all participating clients are required to initiate iterations from the latest global model in each round of Local-SGD, the overall training process can be slowed down due to the *straggler effect*. To address this issue, we propose a *Delayed Local-SGD (DLSGD)* framework for distributed and federated learning with partial client participation. In DLSGD, each client performs local training starting from outdated models, regardless of whether it participates in the global aggregation. We investigate two types of DLSGD methods applied to scenarios where clients have identical or different local objective functions. Theoretical analyses demonstrate that DLSGD achieves asymptotic convergence rates that are on par with the classic Local-SGD methods for solving nonconvex problems, and guarantees *linear speedup* with respect to the number of participating clients. Additionally, we carry out numerical experiments using real datasets to validate the efficiency and scalability of our approach when training neural networks.

## 1 INTRODUCTION

We consider the distributed learning scenario where multiple clients (such as CPUs, GPUs, wireless sensors, and mobile devices) collaboratively train a global model under the orchestration of a central server. This objective can be cast as solving the following optimization problem:

$$\min_{\boldsymbol{w} \in \mathbb{R}^d} F(\boldsymbol{w}) := \frac{1}{N} \sum_{i=1}^{N} F_i(\boldsymbol{w}), \tag{1}$$

where $F_i(\boldsymbol{w}) := \mathbb{E}_{\boldsymbol{\xi}_i \sim \mathcal{D}_i}[f_i(\boldsymbol{w}; \boldsymbol{\xi}_i)]$ is the local objective function and $N$ is the number of clients. Here, $\mathcal{D}_i$ represents the data distribution accessible by client $i$ supported on sample space $\Xi_i$ and $f_i(\cdot; \boldsymbol{\xi}_i) : \mathbb{R}^d \to \mathbb{R}$ is client $i$'s local loss function that is continuously differentiable for any given data sample $\boldsymbol{\xi}_i \in \Xi_i$. In this paper, we focus on the situations in which the $N$ clients can communicate with the central fusion server while there is no peer-to-peer communication among them. Problem 1 encompasses both distributed training (or parallel optimization) (Bertsekas & Tsitsiklis, 2015) and the emerging field of federated learning (FL) that has received immense attention in recent years (Li et al., 2020; Kairouz et al., 2021). In the former case, referred to as the *homogeneous* setting, all clients have identical loss functions $f_i = f_j$ and data distributions $\mathcal{D}_i = \mathcal{D}_j$ for all $i \neq j$. In contrast, the latter case, known as the *heterogeneous* setting, involves clients with distinct local loss functions $f_i$ and data distributions $\mathcal{D}_i$.

The *parallel stochastic gradient descent (parallel SGD)* method solves Problem (1) by allowing for multiple training examples to be processed simultaneously across different clients. However, traditional parallel SGD may incur substantial communication overhead, as each computed stochastic gradient needs to be sent to the server. To alleviate this challenge, *Local-SGD* performs multiple iterations locally at the clients before communicating with the server, which has garnered great interest in the field of distributed/federated learning lately. There exist a batch of Local-SGD variants in the literature, e.g., Mangasarian (1995); Zinkevich et al. (2010); Stich (2019). Here, let us recap a generic version called Generalized FedAvg introduced in Yang et al. (2021), where $n$ out of the $N$

clients start with $\boldsymbol{w}_i^{t,0} = \boldsymbol{w}^t$ and parallelly execute $K$-step SGD during the global round $t$ ($t \geq 0$) according to the following update rule:

$$\boldsymbol{w}_i^{t,k+1} = \boldsymbol{w}_i^{t,k} - \bar{\eta}\nabla f_i(\boldsymbol{w}_i^{t,k};\boldsymbol{\xi}_i^{t,k}) \text{ for } k = 0, 1, \ldots, K-1. \tag{2}$$

In the midst, $\boldsymbol{\xi}_i^{t,k}$ are independently sampled according to distribution $\mathcal{D}_i$ and $\bar{\eta} > 0$ is the local learning rate. Once the $n$ clients finish their local training and upload their last iterates $\boldsymbol{w}_1^{t,K}, \ldots, \boldsymbol{w}_n^{t,K}$, the server proceeds with the global iteration using the following update rule:

$$\boldsymbol{w}^{t+1} = \boldsymbol{w}^t - \frac{\eta}{n}\sum_{i=1}^{n}(\boldsymbol{w}_i^{t,K} - \boldsymbol{w}^t), \tag{3}$$

where $\eta > 0$ is the global learning rate. Note that the Local-SGD procedures (2)–(3) with $\eta = 1$ reduce to the vanilla FedAvg algorithm (McMahan et al., 2017) for FL.

The frequency of communication in Local-SGD can be reduced by allowing each device to process more data locally. Furthermore, it is reported in the literature that performing multiple local iterations can lead to better generalization performance than parallel SGD with large mini-batches (Lin et al., 2020). It has been shown that Local-SGD converges at a rate of $\mathcal{O}(1/\sqrt{nKT})$ for strongly convex objectives in the homogeneous setting (Stich, 2019) and at a rate of $\mathcal{O}(\sqrt{K}/\sqrt{nT})$ for non-convex objectives in the heterogeneous setting (Yang et al., 2021). This indicates that Local-SGD achieves *linear speedup* with respect to (wrt) the number of participating clients $n$, In other words, the number of communication rounds required to achieve a certain level of optimization accuracy decreases proportionally as the number of participating clients increases. Linear speedup is a highly desirable property in distributed learning, since it signifies the scalability and efficiency of the optimization process as more computational resources are added.

In spite of the favorable practical and theoretical properties of Local-SGD, a typical challenge concerns the time needed to synchronize a communication round. Due to the potential diversity of the hardware, there can be large variations in the computation and communication speeds of different clients (Assran et al., 2020; Li et al., 2020). As a consequence, the time needed for a round is conditioned to the slowest client, while the faster clients stay idle once they send their local updates to the server. This phenomenon, known as the *straggler effect*, can significantly slow down the overall training process. Therefore, we are motivated to address the following question: *Can we develop improved Local-SGD methods that effectively mitigate the straggler effect, while still maintaining fast convergence rates and linear speedup guarantees for both the homogeneous and heterogeneous settings?*

## 1.1 OUR CONTRIBUTIONS

In this paper, we provide an affirmative answer to the aforementioned question. Specifically, we propose a *Delayed Local-SGD (DLSGD)* framework, for solving Problem (1) under both the homogeneous (named *DLSGD-homo*) and heterogeneous (named *DLSGD-hetero*) settings. It possesses the following main features:

- *Partial Participation.* Only a subset of clients participate in the global iteration in each round. For DLSGD-homo, the server collects the first $n$ clients' local updates to arrive; while for DLSGD-hetero, the server sample $n$ clients uniformly with replacement.

- *Delayed Local Training.* The clients continuously receiving the latest global models from the server and store them into their *receive buffer* with overwriting permitted. They then perform local training, regardless of their participation in the global iteration. The server accepts delayed updates without waiting for participating clients to train the latest global model in the current round.

- *Asynchronous Clients.* Once a client completes its local training, the delayed update is either uploaded to the server (for DLSGD-homo) or stored in the *send buffer* until it is selected by the server (for DLSGS-hetero). The client then initiates new local training using the global model stored in its receive buffer, independent of the computation/communication states of other clients.

In a nutshell, at the cost of additional storage for local buffers, DLSGD enhances the training efficiency compared to traditional Local-SGD by enabling partial information aggregation, reducing the waiting time at the server, and utilizing the idle time at the clients.

We provide the convergence analyses of DLSGD for solving Problem (1) with *smooth and nonconvex* objective functions, including neural networks that are of particular interest. Our analysis reveals that DLSGD-homo (respectively, DLSGD-hetero) converges at a rate of $\mathcal{O}(1/\sqrt{nKT} + 1/(KT))$ (respectively, $\mathcal{O}(1/\sqrt{nT} + 1/(KT))$) for sufficiently large $T$, which implies a linear speedup wrt to the number of participating clients $n$. Our results demonstrate that incorporating delayed updates in local training effectively mitigates the straggler effect without sacrificing the asymptotic convergence performance. To our knowledge, we are the first to provide convergence guarantees of the asynchronous variants of Local-SGD that exhibit linear speedup. Furthermore, our theoretical findings for DLSGD can be reduced to the latest results for classic Local-SGD and asynchronous SGD under different specific setups, adding to the versatility and applicability of our framework.

We validate the effectiveness of DLSGD through extensive experiments involving the training of neural networks on real-world datasets. We compare its performance with that of the latest distributed learning algorithms, serving as a benchmark for evaluation. These experiments provide empirical evidence for the benefits and advantages of DLSGD in practical scenarios.

## 1.2  RELATED WORKS

**Linear Speedup Analyses of Local-SGD.** The concept of local training has been extensively adopted in distributed and federated learning to reduce the communication cost (Wang et al., 2021). Under the homogeneous setting, the convergence of Local-SGD methods has been studied in Stich (2019); Stich & Karimireddy (2020); Khaled et al. (2020) for (strongly) convex objectives and in Zhou & Cong (2018); Yu et al. (2019b); Stich & Karimireddy (2020) for nonconvex objectives. Under the heterogeneous setting, the convergence rates of Local-SGD has been established in Haddadpour et al. (2019) for objectives satisfying the Polyak-Łojasiewicz condition and in Wang & Joshi (2021); Yang et al. (2021) for general nonconvex objectives. Other variants of Local-SGD include Yu et al. (2019a), which employs momentum techniques to accelerate convergence, and Karimireddy et al. (2020), which incorporates control variates for variance reduction. The recent work by Gu et al. (2021) adopts a strategy that aggregates (possibly stale) stochastic gradients from *all* clients in each round. However, this approach requires the server to wait for active clients to complete local training using the *current* global model. While these algorithms achieve comparable convergence rates with linear speedup with respect to the number of participating clients, they are fully synchronous and rely on (a subset of) clients uploading up-to-date training results. This makes them vulnerable to stragglers and fundamentally different from DLSGD.

**Asynchronous SGD.** There are various asynchronous variants of parallel SGD where clients operate at their own speeds without the need for synchronization, resulting in the server aggregating stale stochastic gradients. Under the homogeneous setting, the linear-speedup convergence of asynchronous SGD has been established in Agarwal & Duchi (2011) for convex objectives, in Lian et al. (2015); Dutta et al. (2018) for nonconvex objectives, and in Arjevani et al. (2020) for convex quadratic objectives. In the heterogeneous setting, Gao et al. (2021) achieves linear speedup by assuming identical staleness of the aggregated gradients. On the other hand, Koloskova et al. (2022) provides the convergence analysis of asynchronous SGD for nonconvex objectives, while the linear speedup is not attained. It is worth noting that, unlike DLSGD, asynchronous SGD typically involves clients solely responsible for gradient computation without performing local training. As a result, asynchronous SGD often requires more frequent communication with the server compared to DLSGD.

**(Semi-)Asynchronous FL.** FL generally concerns data heterogeneity and incorporates local training strategies. According to the taxonomy presented in Xu et al. (2021), our DLSGD-hetero can be categorized as a *semi-asynchronous* FL algorithm since the server's iteration occurs only when a specific subset of clients' delayed updates arrive. A similar algorithm called FedBuff (Nguyen et al., 2022) has demonstrated state-of-the-art empirical performance in semi-asynchronous FL. In recent works such as Zakerinia et al. (2022); Leconte et al. (2023), a different asynchrony model is adopted, where the server can interrupt the clients to request their intermediate local updates. Other works, such as Xie et al. (2019); Chen et al. (2020); Fraboni et al. (2023), enable a higher level of system asynchrony, where the server immediately updates the global model upon receiving a local model from a client. While the existing papers on (semi-)asynchronous FL generally provide convergence analyses on nonconvex objectives, the guarantees for linear speedup are still lacking.

---

**Algorithm 1** DLSGD-homo

| Server executes: | Client $i$ executes: |
|---|---|
| 1: **Input:** $\boldsymbol{w}^0 \in \mathbb{R}^d$, $n, T \in \mathbb{Z}_+$, $\eta > 0$ | 1: **Input:** $K \in \mathbb{Z}_+$, $\bar{\eta} > 0$ |
| 2: Broadcast $\boldsymbol{w}$ to all the clients | 2: **if** the receive buffer is nonempty **then** |
| 3: **for** $t = 0, 1, \ldots, T-1$ **do** | 3:    Retrieve $\boldsymbol{w}$ from the receive buffer |
| 4:    Collect $n$ local updates $\{\Delta_i \ : \ i \in I_t\}$ | 4:    Set $\boldsymbol{w}_i \leftarrow \boldsymbol{w}$ |
|     from any $n$ clients $I_t$ | 5:    **for** $k = 0, 1, \ldots, K-1$ **do** |
| 5:    Set $\Delta_i^t \leftarrow \Delta_i$ | 6:      Sample $\boldsymbol{\xi}_i \sim \mathcal{D}$ |
| 6:    Aggregation: $\Delta^t \leftarrow \frac{1}{n} \sum_{i \in I_t} \Delta_i^t$ | 7:      Set $\boldsymbol{w}_i \leftarrow \boldsymbol{w}_i - \bar{\eta} \nabla f(\boldsymbol{w}_i; \boldsymbol{\xi}_i)$. |
| 7:    Global iteration: $\boldsymbol{w}^{t+1} \leftarrow \boldsymbol{w}^t - \eta \Delta^t$ | 8:    **end for** |
| 8:    Broadcast $\boldsymbol{w}$ to all the $N$ clients | 9:    Send $\Delta_i = \boldsymbol{w} - \boldsymbol{w}_i$ to the server |
| 9: **end for** | 10: **end if** |

---

## 2 ALGORITHM DESCRIPTIONS

In this section, we present the algorithmic details of DLSGD under the homogeneous and heterogeneous settings, respectively.

### 2.1 DLSGD-HOMO

We first introduce DLSGD-homo for solving Problem (1), where for all $i \in [N]$, the local datasets $\mathcal{D}_i$ are set to be identical, denoted as $\mathcal{D}$, where $\mathcal{D}$ is supported on the sample space $\Xi$. Furthermore, the local objective functions $f_i$ are all equal to a continuously differentiable function $f : \mathbb{R}^d \times \Xi \to \mathbb{R}$. Then, the formulation (1) can be simplified as follows:

$$\min_{\boldsymbol{w} \in \mathbb{R}^d} F(\boldsymbol{w}) = \mathbb{E}_{\boldsymbol{\xi} \sim \mathcal{D}}[f(\boldsymbol{w}; \boldsymbol{\xi})]. \tag{4}$$

Problem (4) is applicable to two different distributed scenarios: batch data in a shared memory (Assran et al., 2020) and streaming data from an unknown distribution (Chang et al., 2020). In the former case, the clients (such as CPUs and GPUs) can read simultaneously from a shared dataset, where $\mathcal{D}$ represents the *empirical distribution* supported on the data points, and $\boldsymbol{\xi} \sim \mathcal{D}$ indicates that $\boldsymbol{\xi}$ is uniformly sampled from the dataset. In the latter case, we assume that data are revealed to different clients (such as wireless sensors (De Francisci Morales et al., 2016) and financial sectors (Umadevi et al., 2018)) in an online fashion from a source with the unknown distribution $\mathcal{D}$, and are processed only once without the need to be stored locally.

DLSGD-homo comprises the following main components:

**Warmup Phase.** In the initial round $t = 0$ of DLSGD-homo, the server initialize a global model $\boldsymbol{w}^0$ and broadcast it to the *receive buffers* of all the clients. Each client fetches the global model $\boldsymbol{w}^0$ from their receive buffer and perform parallel local training starting from $\boldsymbol{w}^0$.

**Server's Procedures in Round $t$.** We focus on the situation in which the clients have varying computation/communication speeds. Thus, instead of waiting for all the clients to complete local training and upload their results, the server only needs to cache the first $n$ clients' local updates that arrive. In other words, the clients participate in a non-random manner at their own individual pace, and the server does not interrupt the remaining clients' ongoing work. We let $I_t$ be the set of participating clients with $|I_t| = n$, then the server creates a new global model $\boldsymbol{w}^{t+1}$ (the round index increments by one) as follows:

$$\boldsymbol{w}^{t+1} = \boldsymbol{w}^t - \eta \boldsymbol{g}^t, \tag{5}$$

where $\eta > 0$ is the global learning rate and $\boldsymbol{g}^t := \frac{1}{|I_t|} \sum_{i \in I_t} \Delta_i^t$ and $\Delta_i^t$ is client $i$'s local update that will be specified later. Then, $\boldsymbol{w}^{t+1}$ is broadcast to all the $N$ clients' *receive buffers*.

**Clients' Procedures in Round $t$.** At the beginning of round $t$, it is possible that a client's local training was initiated from a stale version of the global model. Specifically, we use $\tau_i(t)$ to denote the round index of the global model that client $i$ has used in local training in round $t$, then it is obvious that $0 \leq \tau_i(t) \leq t$. Let us assume that client $i$ initiated local training using $\boldsymbol{w}_i^{\tau_i(t),0} = \boldsymbol{w}^{\tau_i(t)}$, where

---

**Algorithm 2** DLSGD-hetero

---

Server executes:
1: **Input:** $\boldsymbol{w}^0 \in \mathbb{R}^d$, $n, T \in \mathbb{Z}_+$, $\eta > 0$
2: Broadcast $\boldsymbol{w}$ to all the clients
3: **for** $t = 0, 1, \dots, T - 1$ **do**
4:    Select a subset $\mathcal{I}_t$ from $[N]$ $n$ independent times using uniform sampling with replacement
5:    Collect the model updates $\{\Delta_i : i \in \mathcal{I}_t\}$ from the clients in $\mathcal{I}_t$
6:    Set $\Delta_i^t \leftarrow \Delta_i$
7:    Aggregation: $\Delta^t \leftarrow \frac{1}{n} \sum_{i \in \mathcal{I}_t} \Delta_i^t$
8:    Global iteration: $\boldsymbol{w}^{t+1} \leftarrow \boldsymbol{w}^t - \eta \Delta^t$
9:    Broadcast $\boldsymbol{w}^t$ to all the $N$ clients
10: **end for**

Client $i$ executes:
1: **Input:** $K \in \mathbb{Z}_+$, $\bar{\eta} > 0$
2: **if** the receive buffer is nonempty **then**
3:    Retrieve $\boldsymbol{w}$ from the receive buffer
4:    Set $\boldsymbol{w}_i \leftarrow \boldsymbol{w}$
5:    **for** $k = 0, 1, \dots, K - 1$ **do**
6:       Sample $\boldsymbol{\xi}_i \sim \mathcal{D}_i$
7:       Set $\boldsymbol{w}_i \leftarrow \boldsymbol{w}_i - \bar{\eta} \nabla f_i(\boldsymbol{w}_i; \boldsymbol{\xi}_i)$.
8:    **end for**
9:    Set $\Delta_i = \boldsymbol{w} - \boldsymbol{w}_i$
10:    Update the send buffer with $\Delta_i$
11: **end if**
12: **if** client $i$ is selected by the server **then**
13:    **while** the send buffer is nonempty **do**
14:       Send $\Delta_i$ to the server
15:    **end while**
16: **end if**

---

$\boldsymbol{w}^{\tau_i(t)}$ represents the global model at round $\tau_i(t)$. Then, the local sequence $\boldsymbol{w}_i^{\tau_i(t),1}, \dots, \boldsymbol{w}_i^{\tau_i(t),K}$ is generated through $K$-step SGD iterations with $K \geq 1$:

$$\boldsymbol{w}_i^{\tau_i(t),k+1} = \boldsymbol{w}_i^{\tau_i(t),k} - \bar{\eta} \nabla f(\boldsymbol{w}_i^{\tau_i(t),k}; \boldsymbol{\xi}_i^{t,k}) \text{ for } k = 0, 1, \dots, K - 1, \qquad (6)$$

where $\bar{\eta} > 0$ is the local learning rate and $\boldsymbol{\xi}_i^{t,k}$'s are independently sampled from $\mathcal{D}$. It is worth emphasizing that we index the data sample $\boldsymbol{\xi}_i^{t,k}$ by $t$ for all $i \in [N]$ and $k = 0, 1, \dots, K - 1$ to highlight that all previous models $\boldsymbol{w}^0, \boldsymbol{w}^1, \dots, \boldsymbol{w}^{t-1}$ do not depend on it. Once the local training of client $i$ is completed, the local update $\Delta_i^t := \boldsymbol{w}_i^{\tau_i(t),0} - \boldsymbol{w}_i^{\tau_i(t),K}$ shall be sent to the server. Then, if the receive buffer is nonempty, the client proceeds with new local training by fetching the latest model from the receive buffer.

The pseudo-code of DLSGD-homo is described in Algorithm 1. We remark that the $N$ clients can work in concurrent with each other as well as with the server. This distinguishes DLSGD-homo from vanilla Local-SGD, where the local iterations (2) and the global iteration (3) are executed alternately. Furthermore, we notice that there is a similar asynchronous Local-SGD algorithm mentioned in Stich (2019, Section 5) for solving Problem (4) with strongly convex objectives. In that algorithm, the local steps $K$ can vary across clients, the local learning rates diminish to 0, and there is no global learning rate. These features also bring about distinct theoretical result compared to that of DLSGD-homo, which will be discussed in Section 3.

## 2.2 DLSGD-HETERO

DLSGD-hetero is designed to address the heterogeneous setting of Problem (1), which falls under the scope of FL. The warm-up phase of DLSGD-hetero remains the same as described in Section 2.1, thus we focus on the key distinctions in terms of the server's and the clients' procedures.

**Server's Procedures in Round $t$.** When the clients have diverse communication/computation speeds, the non-random/deterministic client participation scheme employed in DLSGD-homo and some other FL algorithms (Li et al., 2019; Nguyen et al., 2022) tends to favor fast clients. This can result in a biased exploitation of the local information in $F_i$, potentially impeding convergence. Therefore, we adopt a random client participation scheme, where the server samples clients uniformly with replacement $n$ independent times in each round. The selected clients upload their local updates once the send buffer is nonempty, while the remaining clients continue their ongoing work. We use $\mathcal{I}_t$ (in calligraphic font) to denote the *random* set of the selected clients in round $t$. Since each client can be selected more than once, $\mathcal{I}_t$ is a multiset that may contain repeated elements and $|\mathcal{I}_t| \leq n$. Once the clients in $\mathcal{I}_t$ have uploaded their local updates $\Delta_i^t$'s, the server proceeds to create a new global model $\boldsymbol{w}^{t+1}$ similar to (5):

$$\boldsymbol{w}^{t+1} = \boldsymbol{w}^t - \eta \boldsymbol{h}^t, \qquad (7)$$

while $\boldsymbol{h}^t = \frac{1}{|\mathcal{I}_t|} \sum_{i \in \mathcal{I}_t} \Delta_i^t$. Then, $\boldsymbol{w}^{t+1}$ is broadcast to all the $N$ clients' *receive buffers*.

**Clients' Procedures in Round** $t$. Analogous to the procedures described in Section 2.1, suppose that client $i$ initiated training with $\boldsymbol{w}_i^{\tau_i(t),0} = \boldsymbol{w}^{\tau_i(t)}$ and the $K$-step SGD iterations with $K \geq 1$:

$$\boldsymbol{w}_i^{\tau_i(t),k+1} = \boldsymbol{w}_i^{\tau_i(t),k} - \bar{\eta} \nabla f_i(\boldsymbol{w}_i^{\tau_i(t),k}; \boldsymbol{\xi}_i^{t,k}) \text{ for } k = 0, 1, \dots, K-1, \tag{8}$$

where $\bar{\eta} > 0$ is the local learning rate and $\boldsymbol{\xi}_i^{t,k}$'s are independently sampled from $\mathcal{D}_i$. The client upload its local update $\Delta_i^t = \boldsymbol{w}_i^{\tau_i(t),0} - \boldsymbol{w}_i^{\tau_i(t),K}$ to the server if it is selected by the server; otherwise, it places $\Delta_i^t$ into to the *send buffer* until the client is selected at a later time. Moreover, the client has the capability to regularly retrieve the most recent global models from its receive buffer. This allows the client to initiate new cycles of local training promptly and overwrite the send buffer. By doing so, the client ensures minimal delay in its local update when it is selected in a subsequent round.

We describe the pseudo-code of DLSGD-hetero in Algorithm 2, in which the clients and the server also work in a parallel manner. When $\tau_i(t) = t$ for all $i \in [N]$, DLSGD-hetero reduces to the Generalized FedAvg with partial client participation (Yang et al., 2021). In this case, the participating clients in round $t$ commence local training using the global model $\boldsymbol{w}^t$, while the remaining clients remain idle and do not contribute to the training process.

## 3 THEORETICAL ANALYSES

Our convergence analyses of DLSGD rely on the following standard assumptions.

**Assumption 1** (*L*-smoothness). *There exists $L > 0$ such that for all $i \in [N]$ and $\boldsymbol{w}, \boldsymbol{w}' \in \mathbb{R}^d$,*

$$\|\nabla F_i(\boldsymbol{w}) - \nabla F_i(\boldsymbol{w}')\|_2 \leq L \|\boldsymbol{w} - \boldsymbol{w}'\|_2.$$

**Assumption 2** (unbiased stochastic gradient). *Suppose that $\boldsymbol{\xi}_i \in \Xi_i$ is a data sample and $\boldsymbol{w} \in \mathbb{R}^d$ is a local/global iterate generated by DLSGD, and $\mathcal{F}$ is a sigma algebra. If $\boldsymbol{w}$ is $\mathcal{F}$-measurable and $\boldsymbol{\xi}_i$ is independent of $\boldsymbol{w}$, then for all $i \in [N]$,*

$$\mathbb{E}[\nabla f_i(\boldsymbol{w}; \boldsymbol{\xi}_i)|\mathcal{F}] = \nabla F_i(\boldsymbol{w}). \tag{9}$$

Note that $\mathbb{E}_{\boldsymbol{\xi} \sim \mathcal{D}_i}[\nabla f_i(\boldsymbol{w}; \boldsymbol{\xi}_i)] \neq \nabla F_i(\boldsymbol{w})$ if $\boldsymbol{w}$ is considered to be random, as the iterate $\boldsymbol{w}$ can be a function of (thus dependent on) $\boldsymbol{\xi}_i$ in general. To ensure the unbiasedness property (9), we need the conditions that $\mathcal{F}$ contains all information in $\boldsymbol{w}$ and $\boldsymbol{\xi}_i$ is independent of $\boldsymbol{w}$. Similarly, we impose the following uniform upper bound condition on the conditional variance of stochastic gradients:

**Assumption 3** (bounded variance). *Suppose that $\boldsymbol{\xi}_i \in \Xi_i$ is a data sample and $\boldsymbol{w} \in \mathbb{R}^d$ is a local/global iterate generated by DLSGD, and $\mathcal{F}$ is a sigma algebra. If $\boldsymbol{w}$ is $\mathcal{F}$-measurable and $\boldsymbol{\xi}_i$ is independent of $\boldsymbol{w}$, then there exists $\sigma > 0$ such that for all $i \in [N]$ and $k = 0, 1, \dots, K-1$,*

$$\mathbb{E}\left[\|\nabla f_i(\boldsymbol{w}_i; \boldsymbol{\xi}_i) - \nabla F_i(\boldsymbol{w}_i)\|_2^2 \mid \mathcal{F}\right] \leq \sigma^2.$$

**Assumption 4** (bounded delay). *There exists an integer $\lambda \geq 1$ such that for all $i \in [N]$ and $t \geq 0$,*

$$\tau_i(t) \geq (t - \lambda)_+,$$

*where $(x)_+ := \max\{x, 0\}$ for $x \in \mathbb{R}$.*

Assumption 4 is common in distributed optimization using stale gradients, e.g., Lian et al. (2015); Nguyen et al. (2022), which indicates that the clients can be arbitrarily delayed as long as they participate in the global iterations at least once in the last $\lambda$ rounds. For Algorithm 1, the value of $\lambda$ is a system attribute that is determined by the computation/communication time of the clients and the server. For Algorithm 2, due to the randomness in client sampling, there is a chance (albeit tiny) that a client is not selected throughout the training process and thus Assumption 4 is violated. Indeed, according to Gu et al. (2021, Theorem 5.2), it holds with probability at least $1 - \delta$ that $\lambda \leq \mathcal{O}\left(\frac{1}{p}\left(1 + \log \frac{NT}{\delta}\right)\right)$, where $p = 1 - \left(\frac{N-1}{N}\right)^n$ is the probability of each client being sampled in a single round. In other words, the maximum delay $\lambda$ of DLSGD-hetero depends logarithmically on $T$ with high probability (rather than deterministically).

### 3.1 CONVERGENCE RATE OF DLSGD-HOMO

Now, we present the following convergence rate of DLSGD-homo:

**Theorem 1.** *Let $F^* \in \mathbb{R}$ be the optimal function value of Problem (4), $\boldsymbol{w}^0 \in \mathbb{R}^d$ be the initial point, and $\{\boldsymbol{w}^t\}_{t \geq 1}$ be the sequence generated by Algorithm 1 for solving Problem (4). Suppose that Assumptions 1–4 hold and the learning rates satisfy*

$$\bar{\eta} \leq \frac{1}{4\sqrt{3}LK} \text{ and } \eta\bar{\eta} \leq \frac{1}{4LK\lambda}. \tag{10}$$

*If $\eta = \sqrt{nK(F(\boldsymbol{w}^0) - F^*)/2}$ and $\bar{\eta} = 1/(\sqrt{\sigma^2 LT}K)$, then it holds for $T \geq 1$ that*

$$\frac{1}{T} \sum_{t=0}^{T-1} \mathbb{E}\|\nabla F(\boldsymbol{w}^t)\|_2^2 \leq \sqrt{\frac{128(F(\boldsymbol{w}^0) - F^*)}{nKT}} + \frac{8L}{\sigma^2 KT}. \tag{11}$$

The proof of Theorem 1 is deferred to Appendix A.1. Note that condition (10) implies a lower bound on the required number of global rounds:

$$T \geq \max\left\{ \frac{48L}{\sigma^2}, \frac{(F^* - F(\boldsymbol{w}^0))8nLK\lambda^2}{\sigma^2} \right\} = \Omega(nK\lambda^2). \tag{12}$$

The convergence bound (11) suggests that the average gradient norm converges to zero at a rate of $\mathcal{O}(1/\sqrt{nKT} + 1/(KT))$, which indicates a linear speedup wrt to the number of participating clients $n$ as $T$ is sufficiently large.

When $\lambda = 1$, indicating no delay in the updates, Theorem 1 aligns with the convergence result for synchronous Local-SGD presented in Stich & Karimireddy (2020), which achieves $\mathcal{O}(1/\sqrt{nKT} + n/T)$ rate provided that $T = \Omega(nK)$. Notably, our result for DLSGD-homo demonstrates that the delay in local training does not hinder the asymptotic convergence rate when compared to synchronous Local-SGD.

When $K = 1$, indicating no local training, the convergence bound (11) simplifies to the result presented in Lian et al. (2015) up to a high-order term. This result demonstrates that the asynchronous SGD converges at a rate of $\mathcal{O}(1/\sqrt{nT})$ when $T = \Omega(n\lambda^2)$. Besides, it is worth noting that condition (11) implies that $K = \mathcal{O}(T/(n\lambda^2))$. Consequently, if the number of local steps $K$ is not excessively large, the communication complexity decreases linearly as $K$ increases. This highlights the advantage of local training in DLSGD-homo, as it reduces the number of communication rounds by a factor of $1/K$ asymptotically while maintaining the same sample complexity as asynchronous SGD.

The asynchronous Local-SGD (Stich, 2019) discussed in Section 2.1 attains $\mathcal{O}(1/nT)$ rate for solving Problem (4). Nevertheless, this result is limited to strongly convex objectives under the assumption of bounded gradients. In contrast, Theorem 1 applies to general nonconvex problems without requiring bounded gradients. This sheds light on the distributed training of nonconvex deep neural networks, showcasing the applicability and effectiveness of DLSGD in the more challenging setting.

### 3.2 CONVERGENCE RATE OF DLSGD-HETERO

To analyze the convergence of DLSGD-hetero, we further assume the following upper bounds on gradient norms and the heterogeneity of local objective functions:

**Assumption 5** (bounded gradient). *Suppose that $\boldsymbol{w} \in \mathbb{R}^d$ is a local/global iterate generated by Algorithm 2. Then, there exists $G \geq 0$ such that for all $i \in [N]$,*

$$\mathbb{E}\|\nabla F_i(\boldsymbol{w})\|_2^2 \leq G^2.$$

**Assumption 6** (bounded heterogeneity). *Suppose that $\boldsymbol{w} \in \mathbb{R}^d$ is a local/global iterate generated by Algorithm 2. Then, there exists $\nu \geq 0$ such that for all $i \in [N]$,*

$$\mathbb{E}\|\nabla F_i(\boldsymbol{w}) - \nabla F(\boldsymbol{w})\|_2^2 \leq \nu^2.$$

Then, we can establish the following convergence rate of DLSGD-hetero:

**Theorem 2.** *Let $F^* \in \mathbb{R}$ be the optimal function value of Problem (1), $\boldsymbol{w}^0 \in \mathbb{R}^d$ be the initial point of Algorithm 2, and $\{\boldsymbol{w}^t\}_{t \geq 1}$ be the sequence generated by Algorithm 2. Suppose that Assumptions 1–6 hold and the learning rates satisfy*

$$\bar{\eta} \leq \frac{1}{8LK} \text{ and } \eta\bar{\eta} \leq \min\left\{\frac{1}{4LK\lambda}, \frac{2\sigma^2 + G^2K}{8\sigma^2 LK\lambda + 4LG^2K^2\lambda^2}\right\}. \tag{13}$$

*If $\eta = \sqrt{4nK(F(\boldsymbol{w}^0) - F^*)}$ and $\bar{\eta} = 1/(\sqrt{(4\sigma^2L + 2LKG^2)T}K)$, then it holds for $T \geq 1$ that*

$$\frac{1}{T}\sum_{t=0}^{T-1} \mathbb{E}\|\nabla F(\boldsymbol{w}^t)\|_2^2 \leq \sqrt{\frac{256(F(\boldsymbol{w}^0) - F^*)(16\sigma^2L + 8LG^2K)}{nKT}} + \frac{(\sigma^2 + 8K\nu^2)8L^2}{(4\sigma^2L + 2LKG^2)KT}. \tag{14}$$

The proof of Theorem 2 is given in Appendix A.2. As a consequence of condition (13), we need $T \geq \Omega(n\lambda^4)$, which exhibits a stronger dependence on $\lambda$ compared to the lower bound (12). According to our proof, this results from the heterogeneity of local objectives that necessitates more stringent choices of learning rates for convergence. Therefore, the bound (14) indicates that the average squared gradient norm converges to zero at a rate of $\mathcal{O}(1/\sqrt{nT} + 1/KT)$ for sufficiently large $T$, which also achieves asymptotic linear speedup wrt to $n$.

Similar to the discussion for Theorem 1, Theorem 2 aligns with the existing result for synchronous Local-SGD for federated learning when we set $\lambda = 1$. Specifically, it has been shown in Yang et al. (2021) that the Generalized FedAvg converges at a rate of $\mathcal{O}(\sqrt{K}/\sqrt{nT} + 1/T)$ when $T = \Omega(nK)$ for solving Problem (1). This observation further highlights that the delayed and non-delayed Local-SGD methods exhibit comparable convergence behaviors as $T$ becomes sufficiently large.

Consider the special case when $K = 1$, then Theorem 2 implies a $\mathcal{O}(1/\sqrt{nT} + 1/T)$ convergence rate for $T = \Omega(n\lambda^4)$. This result, as far as we know, provides the first linear speedup guarantee of asynchronous SGD under the heterogeneous setting, which might be of independent interest. In this context, we notice that Koloskova et al. (2022) has established the best-known convergence rate of $\mathcal{O}(1/\sqrt{T} + 1/T)$ for asynchronous SGD under the same setting without the bounded gradient assumption, while the linear speedup property remains unclear.

Similar to DLSGD-hetero, the FedBuff algorithm (Nguyen et al., 2022) also adopts semi-asynchronous local training for solving Problem (1). Differently, FedBuff caches local updates from any $n$ active clients in each round. Such deterministic client participation scheme favors fast clients and thus may impede convergence under the heterogeneous setting. This issue is also evidenced by the theoretical analysis in Nguyen et al. (2022, Section D), which has to make a rather demanding assumption that all clients participate with equal probability so as to guarantee the convergence rate of $\mathcal{O}(1/\sqrt{KT} + K/(n^2T) + \lambda^2/T)$, without achieving asymptotic linear speedup wrt $n$.

## 4 EXPERIMENTS

We apply DLSGD to train convolutional neural networks with two convolutional layers and two fully connected layers. Specifically, we simulate the collaborative training process of $N = 100$ clients with diverse processing capabilities, and evaluate the algorithm performance on image classification tasks using the FashionMNIST (Xiao et al., 2017) and CIFAR-10 (Krizhevsky et al., 2009) datasets. Under the homogeneous setting (indicated by iid), we implement the Local-SGD (2)–(3) and the AsySG (Lian et al., 2015) for comparisons with DLSGD-homo. Under the heterogeneous setting (indicated by non-iid), we implement the Generalized FedAvg (Yang et al., 2021) and Fed-Buff (Nguyen et al., 2022) for comparisons with DLSGD-hetero. The main experiment results and associated discussions are provided in the following. Detailed descriptions of the experimental setup and additional numerical results are deferred to Appendix B.1.

Figure 3 depicts the convergence of DLSGD and its counterparts over wall-clock time. It is observed that DLSGD consistently demonstrates the most rapid decrease in training losses and increase in test accuracies. In the IID case, Local-SGD is dragged down by the straggler clients due to the synchronization in each round, while AsySG necessitates frequent communication between clients and the server in the absence of local training. In comparison, the performance of DLSGD-homo is enhanced by leveraging both asynchronous updates and local iterations. In the non-iid case,

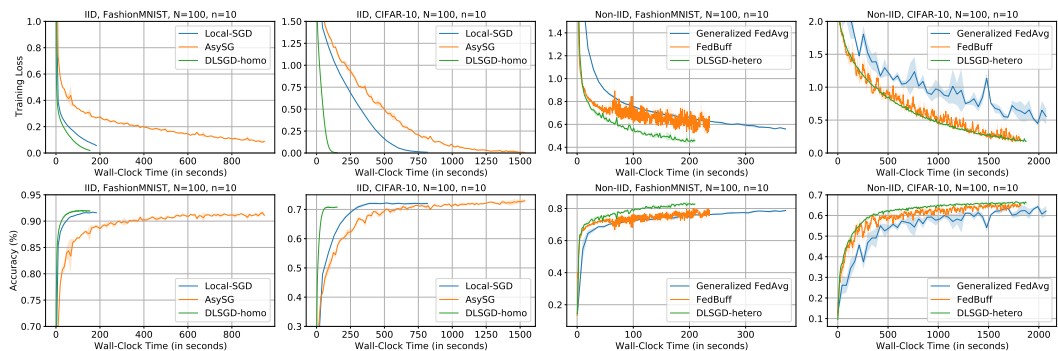

Figure 1: Convergence over wall-clock time of DLSGD and other algorithms with $n = 10$.

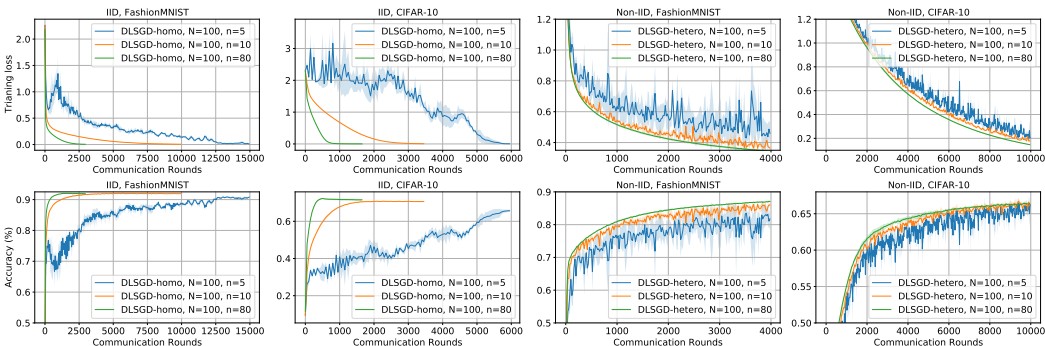

Figure 2: Convergence over communication rounds of DLSGD with different values of $n$.

Generalized FedAvg faces similar challenges due to the presence of straggler clients. On the other hand, while FedBuff allows for semi-asynchronous training, it appears to be more prone to weak generalization. This could be because its training process heavily relies on fast clients, while the datasets at slower clients make a relatively smaller contribution. By contrast, DLSGD-hetero enables uniform client sampling, which helps mitigate the problem by ensuring a balanced contribution from all clients. Additionally, when combined with delayed local training, DLSGD-hetero achieves faster convergence compared to other competitors.

We also explore the impact of the number of participating clients $n$ on the convergence behavior of our proposed DLSGD algorithms. The results, as depicted in Figure 2, demonstrate that increasing the number of participating clients expedites the convergence rates of DLSGD, which supports the convergence results presented in Theorems 1 and 2. The reason behind this phenomenon is that a larger value of $n$ allows for a more accurate estimation of the full gradient during global updates, thereby facilitating faster convergence over the communication rounds. Conversely, when $n$ is small, there is greater variance in the aggregated stochastic gradients, resulting in slower convergence rates and more oscillations in the performance curves.

## 5 CONCLUSION AND DISCUSSION

This paper introduces the DLSGD framework to address the straggler effect in Local-SGD for both homogeneous and heterogeneous distributed learning, by enabling asynchronous local training while utilizing receive/send buffers across clients. Theoretical analyses demonstrate that DLSGD achieves convergence rates comparable to synchronous Local-SGD in the asymptotic regime. Furthermore, it offers linear speedup wrt the number of participating clients, indicating its scalability potential. Numerical experiments on real datasets validate the effectiveness of DLSGD for training neural networks. However, it is worth noting that the lower bound $T = \Omega(nK\lambda^4)$ on the number of training rounds of DLGSD-hetero might have a suboptimal dependence on the maximum delay $\lambda$. Besides, the bounded gradient assumption for Theorem 2 could be an artifact of our analysis. Exploring whether this dependence can be improved or relaxing the assumptions on bounded delay and gradient presents an interesting avenue for further exploration.

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

# A    PROOFS OF THE MAIN RESULTS

We first introduce the two facts that will be frequently used in the proofs.

**Fact 1.** *Let $U$ and $V$ be two random variables, and $\mathcal{F}$ be a sigma algebra. If $U$ is $\mathcal{F}$-measurable, then we have*

$$\mathbb{E}[UV] = \mathbb{E}[U\mathbb{E}[V|\mathcal{F}]].$$

*Proof.* Since $U$ is $\mathcal{F}$-measurable, we have

$$\mathbb{E}[UV|\mathcal{F}] = U\mathbb{E}[V|\mathcal{F}].$$

Then, it follows from the tower rule that

$$\mathbb{E}[UV] = \mathbb{E}[\mathbb{E}[UV|\mathcal{F}]] = \mathbb{E}[U\mathbb{E}[V|\mathcal{F}]],$$

as desired. □

**Fact 2.** *Suppose that $m \geq 2$ is an integer and $\boldsymbol{x}_1, \ldots, \boldsymbol{x}_m$ are vectors in the same inner product space. Then, the following inequality holds:*

$$\left\|\sum_{i=1}^{m} \boldsymbol{x}_i\right\|_2^2 \leq m \sum_{i=1}^{m} \|\boldsymbol{x}_i\|_2^2.$$

## A.1    PROOF OF THEOREM 1

### A.1.1    TECHNICAL LEMMAS

**Lemma 1.** *Suppose that Assumptions 1 and 2 hold. Then, it hold for all $t \geq 0$ that*

$$\mathbb{E}\langle \nabla F(\boldsymbol{w}^t), \bar{\eta}K\nabla F(\boldsymbol{w}^t) - \boldsymbol{g}^t \rangle$$
$$\leq \frac{K}{2}\bar{\eta}\mathbb{E}\|\nabla F(\boldsymbol{w}^t)\|_2^2 + \frac{L^2}{n}\bar{\eta}\sum_{i\in I_t}\sum_{k=0}^{K-1}\mathbb{E}\|\boldsymbol{w}_i^{\tau_i(t),k} - \boldsymbol{w}^{\tau_i(t)}\|_2^2 + \frac{L^2K}{n}\bar{\eta}\sum_{i\in I_t}\mathbb{E}\|\boldsymbol{w}^{\tau_i(t)} - \boldsymbol{w}^t\|_2^2$$
$$- \frac{1}{2K}\bar{\eta}\mathbb{E}\left\|\frac{1}{n}\sum_{i\in I_t}\sum_{k=0}^{K-1}\nabla F(\boldsymbol{w}_i^{\tau_i(t),k})\right\|_2^2.$$

*Proof.* Using the local iteration formula (6) of DLSGD-homo, we have

$$\Delta_i^t = \boldsymbol{w}_i^{\tau_i(t),0} - \boldsymbol{w}_i^{\tau_i(t),K} = \sum_{k=0}^{K-1}\bar{\eta}\nabla f(\boldsymbol{w}_i^{\tau_i(t),k}; \boldsymbol{\xi}_i^{t,k}),$$

which implies that

$$\boldsymbol{g}^t := \frac{1}{|I_t|}\sum_{i\in I_t}\Delta_i^t = \frac{\bar{\eta}}{n}\sum_{i\in I_t}\sum_{k=0}^{K-1}\nabla f(\boldsymbol{w}_i^{\tau_i(t),k}; \boldsymbol{\xi}_i^{t,k}). \tag{15}$$

Therefore, we have

$$\mathbb{E}\langle \nabla F(\boldsymbol{w}^t), \bar{\eta}K\nabla F(\boldsymbol{w}^t) - \boldsymbol{g}^t\rangle$$

$$= \bar{\eta}\mathbb{E}\left\langle \nabla F(\boldsymbol{w}^t), K\nabla F(\boldsymbol{w}^t) - \frac{1}{n}\sum_{i \in I_t}\sum_{k=0}^{K-1}\nabla f(\boldsymbol{w}_i^{\tau_i(t),k}; \boldsymbol{\xi}_i^{t,k})\right\rangle$$

$$= \bar{\eta}\mathbb{E}\left\langle \sqrt{K}\nabla F(\boldsymbol{w}^t), \frac{1}{\sqrt{K}n}\sum_{i \in I_t}\left(K\nabla F(\boldsymbol{w}^t) - \sum_{k=0}^{K-1}\nabla f(\boldsymbol{w}_i^{\tau_i(t),k}; \boldsymbol{\xi}_i^{t,k})\right)\right\rangle$$

$$\overset{(a)}{=} \bar{\eta}\mathbb{E}\left\langle \sqrt{K}\nabla F(\boldsymbol{w}^t), \frac{1}{\sqrt{K}n}\sum_{i \in I_t}\left(K\nabla F(\boldsymbol{w}^t) - \sum_{k=0}^{K-1}\mathbb{E}\left[\nabla f(\boldsymbol{w}_i^{\tau_i(t),k}; \boldsymbol{\xi}_i^{t,k}) \,\Big|\, \boldsymbol{w}^t, \boldsymbol{w}_i^{\tau_i(t),k}\right]\right)\right\rangle,$$

$$\overset{(b)}{=} \bar{\eta}\mathbb{E}\left\langle \sqrt{K}\nabla F(\boldsymbol{w}^t), \frac{1}{\sqrt{K}n}\sum_{i \in I_t}\left(K\nabla F(\boldsymbol{w}^t) - \sum_{k=0}^{K-1}\nabla F(\boldsymbol{w}_i^{\tau_i(t),k})\right)\right\rangle,$$

$$\overset{(c)}{=} \frac{K}{2}\bar{\eta}\mathbb{E}\|\nabla F(\boldsymbol{w}^t)\|_2^2 + \frac{1}{2K}\bar{\eta}\underbrace{\mathbb{E}\left\|\frac{1}{n}\sum_{i \in I_t}\sum_{k=0}^{K-1}\left(\nabla F(\boldsymbol{w}_i^{\tau_i(t),k}) - \nabla F(\boldsymbol{w}^t)\right)\right\|_2^2}_{X^t}$$

$$- \frac{1}{2K}\bar{\eta}\mathbb{E}\left\|\frac{1}{n}\sum_{i \in I_t}\sum_{k=0}^{K-1}\nabla F(\boldsymbol{w}_i^{\tau_i(t),k})\right\|_2^2, \tag{16}$$

where $(a)$ is implied by Fact 1, $(b)$ follows from Assumption 2, and $(c)$ uses the identity $\langle \boldsymbol{x}, \boldsymbol{y}\rangle = (\|\boldsymbol{x}\|_2^2 + \|\boldsymbol{y}\|_2^2 - \|\boldsymbol{x} - \boldsymbol{y}\|_2^2)/2$ for vectors $\boldsymbol{x}$ and $\boldsymbol{y}$. To upper bound $X^t$, we have

$$X^t = \mathbb{E}\left\|\frac{1}{n}\sum_{i \in I_t}\sum_{k=0}^{K-1}\left(\nabla F(\boldsymbol{w}_i^{\tau_i(t),k}) - \nabla F(\boldsymbol{w}^{\tau_i(t)}) + \nabla F(\boldsymbol{w}^{\tau_i(t)}) - \nabla F(\boldsymbol{w}^t)\right)\right\|_2^2$$

$$\overset{(a)}{\leq} 2\mathbb{E}\left\|\frac{1}{n}\sum_{i \in I_t}\sum_{k=0}^{K-1}\left(\nabla F(\boldsymbol{w}_i^{\tau_i(t),k}) - \nabla F(\boldsymbol{w}^{\tau_i(t)})\right)\right\|_2^2 + 2\mathbb{E}\left\|\frac{1}{n}\sum_{i \in I_t}\sum_{k=0}^{K-1}\left(\nabla F(\boldsymbol{w}^{\tau_i(t)}) - \nabla F(\boldsymbol{w}^t)\right)\right\|_2^2$$

$$\overset{(b)}{\leq} \frac{2}{n^2}\sum_{i \in I_t}\sum_{k=0}^{K-1}nK\mathbb{E}\|\nabla F(\boldsymbol{w}_i^{\tau_i(t),k}) - \nabla F(\boldsymbol{w}^{\tau_i(t)})\|_2^2 + \frac{2}{n^2}\sum_{i \in I_t}\sum_{k=0}^{K-1}nK\mathbb{E}\|\nabla F(\boldsymbol{w}^{\tau_i(t)}) - \nabla F(\boldsymbol{w}^t)\|_2^2$$

$$\overset{(c)}{\leq} \frac{2L^2K}{n}\sum_{i \in I_t}\sum_{k=0}^{K-1}\mathbb{E}\|\boldsymbol{w}_i^{\tau_i(t),k} - \boldsymbol{w}^{\tau_i(t)}\|_2^2 + \frac{2L^2K^2}{n}\sum_{i \in I_t}\mathbb{E}\|\boldsymbol{w}^{\tau_i(t)} - \boldsymbol{w}^t\|_2^2, \tag{17}$$

where $(a)$ and $(b)$ follow from Fact 2 with $m = 2$ and $m = nK$, respectively, and $(c)$ is implied by Assumption 1. Plugging (17) back into (16) gives the desired result. $\square$

**Lemma 2.** *Suppose that Assumptions 2, 3, and 4 hold. Then, it holds for all $t \geq 0$ that*

$$\mathbb{E}\left\|\sum_{j \in I_t}\sum_{k=0}^{K-1}\left(\nabla f(\boldsymbol{w}_j^{\tau_j(t),k}; \boldsymbol{\xi}_j^{t,k}) - \nabla F(\boldsymbol{w}_j^{\tau_j(t),k})\right)\right\|_2^2 \leq nK\sigma^2, \tag{18}$$

*Besides, it holds for all $t \geq 1$ and $i \in I_t$ that*

$$\mathbb{E}\left\|\sum_{s=\tau_i(t)}^{t-1}\sum_{j \in I_s}\sum_{k=0}^{K-1}\left(\nabla f(\boldsymbol{w}_j^{\tau_j(s),k}; \boldsymbol{\xi}_j^{s,k}) - \nabla F(\boldsymbol{w}_j^{\tau_j(s),k})\right)\right\|_2^2 \leq nK\lambda\sigma^2. \tag{19}$$

*Proof.* For any integers $i, j \in I_t$ such that $i \neq j$ and $k, \ell \in [0, K-1]$, we have

$$\mathbb{E}\left\langle \nabla f(\boldsymbol{w}_i^{\tau_i(t),k}; \boldsymbol{\xi}_i^{t,k}) - \nabla F(\boldsymbol{w}_i^{\tau_i(t),\ell}), \nabla f(\boldsymbol{w}_j^{\tau_j(t),\ell}; \boldsymbol{\xi}_j^{t,\ell}) - \nabla F(\boldsymbol{w}_j^{\tau_j(t),\ell})\right\rangle$$

$$= \mathbb{E}\left\langle \nabla f(\boldsymbol{w}_i^{\tau_i(t),k}; \boldsymbol{\xi}_i^{t,k}) - \nabla F(\boldsymbol{w}_i^{\tau_i(t),\ell}), \mathbb{E}\left[\nabla f(\boldsymbol{w}_j^{\tau_j(t),\ell}; \boldsymbol{\xi}_j^{t,\ell}) - \nabla F(\boldsymbol{w}_j^{\tau_j(t),\ell}) \,\Big|\, \boldsymbol{w}_j^{\tau_j(t),\ell}, \boldsymbol{w}_i^{\tau_i(t),k}, \boldsymbol{\xi}_i^{t,k}\right]\right\rangle$$

$$= 0,$$

where the first equality uses Fact 1 and the second equality follows from Assumption 2. Therefore,

$$
\mathbb{E} \left\langle \sum_{k=0}^{K-1} \left( \nabla f(\boldsymbol{w}_i^{\tau_i(t),k}; \boldsymbol{\xi}_i^{t,k}) - \nabla F(\boldsymbol{w}_i^{\tau_i(t),k}) \right), \sum_{k=0}^{K-1} \left( \nabla f(\boldsymbol{w}_j^{\tau_j(t),k}; \boldsymbol{\xi}_j^{t,k}) - \nabla F(\boldsymbol{w}_j^{\tau_j(t),k}) \right) \right\rangle
$$
$$
= \sum_{0 \le k,\ell \le K-1} \mathbb{E} \left\langle \nabla f(\boldsymbol{w}_i^{\tau_i(t),k}; \boldsymbol{\xi}_i^{t,k}) - \nabla F(\boldsymbol{w}_i^{\tau_i(t),k}), \nabla f(\boldsymbol{w}_j^{\tau_j(t),\ell}; \boldsymbol{\xi}_j^{t,\ell}) - \nabla F(\boldsymbol{w}_j^{\tau_j(t),\ell}) \right\rangle
$$
$$
= 0. \tag{20}
$$

Therefore, we have

$$
\mathbb{E} \left\| \sum_{j \in I_t} \sum_{k=0}^{K-1} \left( \nabla f(\boldsymbol{w}_j^{\tau_j(t),k}; \boldsymbol{\xi}_j^{t,k}) - \nabla F(\boldsymbol{w}_j^{\tau_j(t),k}) \right) \right\|_2^2
$$
$$
= \sum_{i,j \in I_t} \mathbb{E} \left\langle \sum_{k=0}^{K-1} \left( \nabla f(\boldsymbol{w}_i^{\tau_i(t),k}; \boldsymbol{\xi}_i^{t,k}) - \nabla F(\boldsymbol{w}_i^{\tau_i(t),k}) \right), \sum_{k=0}^{K-1} \left( \nabla f(\boldsymbol{w}_j^{\tau_j(t),k}; \boldsymbol{\xi}_j^{t,k}) - \nabla F(\boldsymbol{w}_j^{\tau_j(t),k}) \right) \right\rangle
$$
$$
= \sum_{j \in I_t} \mathbb{E} \left\| \sum_{k=0}^{K-1} \left( \nabla f(\boldsymbol{w}_j^{\tau_j(t),k}; \boldsymbol{\xi}_j^{t,k}) - \nabla F(\boldsymbol{w}_j^{\tau_j(t),k}) \right) \right\|_2^2. \tag{21}
$$

To proceed, we note that for any integers $k, \ell \in [0, K-1]$ such that $k < \ell$, it follows from Fact 1 and Assumption 2 that

$$
\mathbb{E} \left\langle \nabla f(\boldsymbol{w}_j^{\tau_j(t),k}; \boldsymbol{\xi}_j^{t,k}) - \nabla F(\boldsymbol{w}_j^{\tau_j(t),k}), \nabla f(\boldsymbol{w}_j^{\tau_j(t),\ell}; \boldsymbol{\xi}_j^{t,\ell}) - \nabla F(\boldsymbol{w}_j^{\tau_j(t),\ell}) \right\rangle
$$
$$
= \mathbb{E} \left\langle \nabla f(\boldsymbol{w}_j^{\tau_j(t),k}; \boldsymbol{\xi}_j^{t,k}) - \nabla F(\boldsymbol{w}_j^{\tau_j(t),k}), \mathbb{E} \left[ \nabla f(\boldsymbol{w}_j^{\tau_j(t),\ell}; \boldsymbol{\xi}_j^{t,\ell}) - \nabla F(\boldsymbol{w}_j^{\tau_j(t),\ell}) \,\Big|\, \boldsymbol{w}_j^{\tau_j(t),k}, \boldsymbol{\xi}_j^{t,k} \right] \right\rangle
$$
$$
= 0. \tag{22}
$$

This, together with (21), implies that

$$
\mathbb{E} \left\| \sum_{k=0}^{K-1} \left( \nabla f(\boldsymbol{w}_j^{\tau_j(t),k}; \boldsymbol{\xi}_j^{t,k}) - \nabla F(\boldsymbol{w}_j^{\tau_j(t),k}) \right) \right\|_2^2
$$
$$
= \sum_{0 \le k,\ell \le K-1} \mathbb{E} \left\langle \nabla f(\boldsymbol{w}_j^{\tau_j(t),k}; \boldsymbol{\xi}_j^{t,k}) - \nabla F(\boldsymbol{w}_j^{\tau_j(t),k}), \nabla f(\boldsymbol{w}_j^{\tau_j(t),\ell}; \boldsymbol{\xi}_j^{t,\ell}) - \nabla F(\boldsymbol{w}_j^{\tau_j(t),\ell}) \right\rangle
$$
$$
= \sum_{k=0}^{K-1} \mathbb{E} \left\| \nabla f(\boldsymbol{w}_j^{\tau_j(t),k}; \boldsymbol{\xi}_j^{t,k}) - \nabla F(\boldsymbol{w}_j^{\tau_j(t),k}) \right\|_2^2. \tag{23}
$$

Plugging (23) back into (21) and using Assumption 3 yield

$$
\mathbb{E} \left\| \sum_{j \in I_t} \sum_{k=0}^{K-1} \left( \nabla f(\boldsymbol{w}_j^{\tau_j(t),k}; \boldsymbol{\xi}_j^{t,k}) - \nabla F(\boldsymbol{w}_j^{\tau_j(t),k}) \right) \right\|_2^2
$$
$$
= \sum_{j \in I_t} \sum_{k=0}^{K-1} \mathbb{E} \left\| \nabla f(\boldsymbol{w}_j^{\tau_j(t),k}; \boldsymbol{\xi}_j^{t,k}) - \nabla F(\boldsymbol{w}_j^{\tau_j(t),k}) \right\|_2^2
$$
$$
\le nK\sigma^2, \tag{24}
$$

which completes the proof of (18).

We further note that for any integers $s, r \in [\tau_i(t), t-1]$ such that $s < r$, it follows from Fact 1 and Assumption 2 that

$$
\mathbb{E} \left\langle \sum_{j \in I_s} \sum_{k=0}^{K-1} \left( \nabla f(\boldsymbol{w}_j^{\tau_j(s),k}; \boldsymbol{\xi}_j^{s,k}) - \nabla F(\boldsymbol{w}_j^{\tau_j(s),k}) \right), \sum_{j \in I_s} \sum_{k=0}^{K-1} \left( \nabla f(\boldsymbol{w}_j^{\tau_j(r),k}; \boldsymbol{\xi}_j^{r,k}) - \nabla F(\boldsymbol{w}_j^{\tau_j(r),k}) \right) \right\rangle
$$
$$
= \sum_{i,j \in I_t} \sum_{0 \le k,\ell \le K-1} \mathbb{E} \left\langle \nabla f(\boldsymbol{w}_i^{\tau_i(s),k}; \boldsymbol{\xi}_i^{s,k}) - \nabla F(\boldsymbol{w}_i^{\tau_i(s),k}), \mathbb{E} \left[ \nabla f(\boldsymbol{w}_j^{\tau_j(r),\ell}; \boldsymbol{\xi}_j^{r,\ell}) - \nabla F(\boldsymbol{w}_j^{\tau_j(r),\ell}) \,\Big|\, \boldsymbol{w}_i^{\tau_i(s),k}, \boldsymbol{\xi}_i^{s,k} \right] \right\rangle
$$
$$
= 0.
$$

This implies that

$$
\mathbb{E}\left\|\sum_{s=\tau_i(t)}^{t-1}\sum_{j\in I_s}\sum_{k=0}^{K-1}\left(\nabla f(\boldsymbol{w}_j^{\tau_j(s),k};\boldsymbol{\xi}_j^{s,k})-\nabla F(\boldsymbol{w}_j^{\tau_j(s),k})\right)\right\|_2^2
$$

$$
=\sum_{\tau_i(t)\leq s,r\leq t-1}\mathbb{E}\left\langle\sum_{j\in I_s}\sum_{k=0}^{K-1}\left(\nabla f(\boldsymbol{w}_j^{\tau_j(s),k};\boldsymbol{\xi}_j^{s,k})-\nabla F(\boldsymbol{w}_j^{\tau_j(s),k})\right),\sum_{j\in I_s}\sum_{k=0}^{K-1}\left(\nabla f(\boldsymbol{w}_j^{\tau_j(r),k};\boldsymbol{\xi}_j^{r,k})-\nabla F(\boldsymbol{w}_j^{\tau_j(r),k})\right)\right\rangle
$$

$$
=\sum_{s=\tau_i(t)}^{t-1}\mathbb{E}\left\|\sum_{j\in I_s}\sum_{k=0}^{K-1}\left(\nabla f(\boldsymbol{w}_j^{\tau_j(s),k};\boldsymbol{\xi}_j^{s,k})-\nabla F(\boldsymbol{w}_j^{\tau_j(s),k})\right)\right\|_2^2.
$$

$$
\leq\sum_{s=(t-\lambda)_+}^{t-1}nK\sigma^2
$$

$$
\leq nK\lambda\sigma^2,
$$

where the first inequality uses (24) and the second inequality follows from Assumption 4. This completes the proof of (19). □

**Lemma 3.** *Suppose that Assumptions 2, 3, and 4 hold. Then, it hold for all $t\geq 0$ that*

$$
\mathbb{E}\|\boldsymbol{g}^t\|_2^2\leq 2K\bar{\eta}^2\frac{\sigma^2}{n}+2\bar{\eta}^2\mathbb{E}\left\|\frac{1}{n}\sum_{i\in I_t}\sum_{k=0}^{K-1}\nabla F(\boldsymbol{w}_i^{\tau_i(t),k})\right\|_2^2. \tag{25}
$$

*Proof.* In view of (15), we have

$$
\mathbb{E}\|\boldsymbol{g}^t\|_2^2
$$

$$
=\mathbb{E}\left\|\frac{1}{n}\sum_{i\in I_t}\sum_{k=0}^{K-1}\bar{\eta}\nabla f(\boldsymbol{w}_i^{\tau_i(t),k};\boldsymbol{\xi}_i^{t,k})\right\|_2^2
$$

$$
=\mathbb{E}\left\|\frac{1}{n}\sum_{i\in I_t}\sum_{k=0}^{K-1}\bar{\eta}\left(\nabla f(\boldsymbol{w}_i^{\tau_i(t),k};\boldsymbol{\xi}_i^{t,k})-\nabla F(\boldsymbol{w}_i^{\tau_i(t),k})+\nabla F(\boldsymbol{w}_i^{\tau_i(t),k})\right)\right\|_2^2
$$

$$
\overset{(a)}{\leq}2\mathbb{E}\left\|\frac{1}{n}\sum_{i\in I_t}\sum_{k=0}^{K-1}\bar{\eta}\left(\nabla f(\boldsymbol{w}_i^{\tau_i(t),k};\boldsymbol{\xi}_i^{t,k})-\nabla F(\boldsymbol{w}_i^{\tau_i(t),k})\right)\right\|_2^2+2\mathbb{E}\left\|\frac{1}{n}\sum_{i\in I_t}\sum_{k=0}^{K-1}\bar{\eta}\nabla F(\boldsymbol{w}_i^{\tau_i(t),k})\right\|_2^2
$$

$$
\overset{(b)}{\leq}2K\bar{\eta}^2\frac{\sigma^2}{n}+2\bar{\eta}^2\mathbb{E}\left\|\frac{1}{n}\sum_{i\in I_t}\sum_{k=0}^{K-1}\nabla F(\boldsymbol{w}_i^{\tau_i(t),k})\right\|_2^2. \tag{26}
$$

where the $(a)$ uses Fact 2 with $m=2$, $(b)$ follows from (18) in Lemma 2. □

**Lemma 4.** *Suppose that Assumption 1, 2, and 3 hold. Then, it holds for all $t\geq 0$ that*

$$
\frac{1}{n}\sum_{i\in I_t}\sum_{k=0}^{K-1}\mathbb{E}\|\boldsymbol{w}_i^{\tau_i(t),k}-\boldsymbol{w}^{\tau_i(t)}\|_2^2
$$

$$
\leq 2K^2\bar{\eta}^2\sigma^2+12K^3\bar{\eta}^2\mathbb{E}\|\nabla F(\boldsymbol{w}^t)\|_2^2+\frac{12L^2K^3}{n}\bar{\eta}^2\sum_{i\in I_t}\mathbb{E}\|\boldsymbol{w}^{\tau_i(t)}-\boldsymbol{w}^t\|_2^2.
$$

*Proof.* Using the local iteration formula, we have

$$
\mathbb{E}\|\boldsymbol{w}_i^{\tau_i(t),k} - \boldsymbol{w}^{\tau_i(t)}\|_2^2
$$
$$
= \mathbb{E}\|\boldsymbol{w}_i^{\tau_i(t),k-1} - \bar{\eta}\nabla f(\boldsymbol{w}_i^{\tau_i(t),k-1}; \boldsymbol{\xi}_i^{t,k-1}) - \boldsymbol{w}^{\tau_i(t)}\|_2^2
$$
$$
= \mathbb{E}\Big\|\boldsymbol{w}_i^{\tau_i(t),k-1} - \boldsymbol{w}^{\tau_i(t)} - \bar{\eta}\Big(\nabla F(\boldsymbol{w}_i^{\tau_i(t),k-1}) - \nabla F(\boldsymbol{w}^{\tau_i(t)}) + \nabla F(\boldsymbol{w}^{\tau_i(t)}) - \nabla F(\boldsymbol{w}^t) + \nabla F(\boldsymbol{w}^t)\Big)
$$
$$
\quad - \bar{\eta}\Big(\nabla f(\boldsymbol{w}_i^{\tau_i(t),k-1}; \boldsymbol{\xi}_i^{t,k-1}) - \nabla F(\boldsymbol{w}_i^{\tau_i(t),k-1})\Big)\Big\|_2^2
$$
$$
= \mathbb{E}\Big\|\boldsymbol{w}_i^{\tau_i(t),k-1} - \boldsymbol{w}^{\tau_i(t)} - \bar{\eta}\Big(\nabla F(\boldsymbol{w}_i^{\tau_i(t),k-1}) - \nabla F(\boldsymbol{w}^{\tau_i(t)}) + \nabla F(\boldsymbol{w}^{\tau_i(t)}) - \nabla F(\boldsymbol{w}^t) + \nabla F(\boldsymbol{w}^t)\Big)\Big\|_2^2
$$
$$
\quad + \bar{\eta}^2\mathbb{E}\|\nabla f(\boldsymbol{w}_i^{\tau_i(t),k-1}; \boldsymbol{\xi}_i^{t,k-1}) - \nabla F(\boldsymbol{w}_i^{\tau_i(t),k-1})\|_2^2,
$$

where the last equality holds because Fact 1 and Assumption 2 imply that the cross term is 0, i.e.,

$$
2\mathbb{E}\Big\langle \boldsymbol{w}_i^{\tau_i(t),k-1} - \boldsymbol{w}^{\tau_i(t)} - \bar{\eta}\Big(\nabla F(\boldsymbol{w}_i^{\tau_i(t),k-1}) - \nabla F(\boldsymbol{w}^{\tau_i(t)}) + \nabla F(\boldsymbol{w}^{\tau_i(t)}) - \nabla F(\boldsymbol{w}^t) + \nabla F(\boldsymbol{w}^t)\Big),
$$
$$
\nabla f(\boldsymbol{w}_i^{\tau_i(t),k-1}; \boldsymbol{\xi}_i^{t,k-1}) - \nabla F(\boldsymbol{w}_i^{\tau_i(t),k-1})\Big\rangle
$$
$$
= 2\mathbb{E}\Big\langle \boldsymbol{w}_i^{\tau_i(t),k-1} - \boldsymbol{w}^{\tau_i(t)} - \bar{\eta}\Big(\nabla F(\boldsymbol{w}_i^{\tau_i(t),k-1}) - \nabla F(\boldsymbol{w}^{\tau_i(t)}) + \nabla F(\boldsymbol{w}^{\tau_i(t)}) - \nabla F(\boldsymbol{w}^t) + \nabla F(\boldsymbol{w}^t)\Big),
$$
$$
\mathbb{E}\Big[\nabla f(\boldsymbol{w}_i^{\tau_i(t),k-1}; \boldsymbol{\xi}_i^{t,k-1}) - \nabla F(\boldsymbol{w}_i^{\tau_i(t),k-1}) \,\Big|\, \boldsymbol{w}_i^{\tau_i(t),k-1}, \boldsymbol{w}^t\Big]\Big\rangle
$$
$$
= 0. \tag{27}
$$

It follows that

$$
\mathbb{E}\|\boldsymbol{w}_i^{\tau_i(t),k} - \boldsymbol{w}^{\tau_i(t)}\|_2^2
$$
$$
\overset{(a)}{\leq} \bar{\eta}^2\sigma^2 + \left(1 + \frac{1}{2K-1}\right)\mathbb{E}\|\boldsymbol{w}_i^{\tau_i(t),k-1} - \boldsymbol{w}^{\tau_i(t)}\|_2^2
$$
$$
\quad + 2K\bar{\eta}^2\mathbb{E}\|\nabla F(\boldsymbol{w}_i^{\tau_i(t),k-1}) - \nabla F(\boldsymbol{w}^{\tau_i(t)}) + \nabla F(\boldsymbol{w}^{\tau_i(t)}) - \nabla F(\boldsymbol{w}^t) + \nabla F(\boldsymbol{w}^t)\|_2^2
$$
$$
\overset{(b)}{\leq} \left(1 + \frac{1}{2K-1}\right)\mathbb{E}\|\boldsymbol{w}_i^{\tau_i(t),k-1} - \boldsymbol{w}^{\tau_i(t)}\|_2^2 + 6K\bar{\eta}^2\mathbb{E}\|\nabla F(\boldsymbol{w}_i^{\tau_i(t),k-1}) - \nabla F(\boldsymbol{w}^{\tau_i(t)})\|_2^2
$$
$$
\quad + 6K\bar{\eta}^2\mathbb{E}\|\nabla F(\boldsymbol{w}^{\tau_i(t)}) - \nabla F(\boldsymbol{w}^t)\|_2^2 + 6K\bar{\eta}^2\mathbb{E}\|\nabla F(\boldsymbol{w}^t)\|_2^2 + \bar{\eta}^2\sigma^2
$$
$$
\overset{(c)}{\leq} \left(1 + \frac{1}{2K-1}\right)\mathbb{E}\|\boldsymbol{w}_i^{\tau_i(t),k-1} - \boldsymbol{w}^{\tau_i(t)}\|_2^2 + 6L^2K\bar{\eta}^2\mathbb{E}\|\boldsymbol{w}_i^{\tau_i(t),k-1} - \boldsymbol{w}^{\tau_i(t)}\|_2^2
$$
$$
\quad + 6L^2K\bar{\eta}^2\mathbb{E}\|\boldsymbol{w}^{\tau_i(t)} - \boldsymbol{w}^t\|_2^2 + 6K\bar{\eta}^2\mathbb{E}\|\nabla F(\boldsymbol{w}^t)\|_2^2 + \bar{\eta}^2\sigma^2
$$
$$
\overset{(d)}{\leq} \left(1 + \frac{1}{K-1}\right)\mathbb{E}\|\boldsymbol{w}_i^{\tau_i(t),k-1} - \boldsymbol{w}^{\tau_i(t)}\|_2^2
$$
$$
\quad + 6L^2K\bar{\eta}^2\mathbb{E}\|\boldsymbol{w}^{\tau_i(t)} - \boldsymbol{w}^t\|_2^2 + 6K\bar{\eta}^2\mathbb{E}\|\nabla F(\boldsymbol{w}^t)\|_2^2 + \bar{\eta}^2\sigma^2, \tag{28}
$$

where $(a)$ uses Assumption 3 and the fact that $\|\boldsymbol{x} + \boldsymbol{y}\|_2^2 \leq (1 + 1/\beta)\|\boldsymbol{x}\|_2^2 + (1 + \beta)\|\boldsymbol{y}\|_2^2$ for vectors $\boldsymbol{x}, \boldsymbol{y}$ and $\beta = 2K - 1$, $(b)$ uses Fact 2 with $n = 3$, $(c)$ is implied by Assumption 1, and $(d)$ holds because

$$
\bar{\eta} \leq \frac{1}{2\sqrt{2}LK} \Rightarrow 1 + \frac{1}{2K-1} + 6L^2K\bar{\eta}^2 \leq 1 + \frac{1}{K-1}
$$

Let $Z_i^{t,k} := \mathbb{E}\|\boldsymbol{w}_i^{\tau_i(t),k} - \boldsymbol{w}^{\tau_i(t)}\|_2^2$, $a := 1 + \frac{1}{K-1}$, and $b := 6L^2K\bar{\eta}^2\mathbb{E}\|\boldsymbol{w}^{\tau_i(t)} - \boldsymbol{w}^t\|_2^2 + 6K\bar{\eta}^2\mathbb{E}\|\nabla F(\boldsymbol{w}^t)\|_2^2 + \bar{\eta}^2\sigma^2$, then (28) can be written as

$$
Z_i^{t,k} \leq aZ_i^{t,k-1} + b.
$$

Solving this recursion gives

$$Z_i^{t,k} \leq a^k Z_i^{t,0} + b \sum_{\ell=0}^{k-1} a^\ell$$

$$= \frac{a^k - 1}{a - 1} b$$

$$= (K - 1) \left( \left( 1 + \frac{1}{K-1} \right)^{K-1} - 1 \right) \left( 6L^2 K \bar{\eta}^2 \mathbb{E} \| \boldsymbol{w}^{\tau_i(t)} - \boldsymbol{w}^t \|_2^2 + 6K \bar{\eta}^2 \mathbb{E} \| \nabla F(\boldsymbol{w}^t) \|_2^2 + \bar{\eta}^2 \sigma^2 \right)$$

$$\leq 2(K - 1) \left( \sigma^2 \bar{\eta}^2 + 6K \bar{\eta}^2 \mathbb{E} \| \nabla F(\boldsymbol{w}^t) \|_2^2 + 6L^2 K \bar{\eta}^2 \mathbb{E} \| \boldsymbol{w}^{\tau_i(t)} - \boldsymbol{w}^t \|_2^2 \right),$$

where the second inequality holds due to the fact that $\left( 1 + \frac{1}{K-1} \right)^{K-1} \leq e \leq 3$ with $e$ is being Euler's number. Thus, we obtain

$$\frac{1}{n} \sum_{i \in I_t} \sum_{k=0}^{K-1} Z_i^{t,k}$$

$$\leq \frac{1}{n} \sum_{i \in I_t} \sum_{k=0}^{K-1} 2(K - 1) \left( \sigma^2 \bar{\eta}^2 + 6K \bar{\eta}^2 \mathbb{E} \| \nabla F(\boldsymbol{w}^t) \|_2^2 + 6L^2 K \bar{\eta}^2 \mathbb{E} \| \boldsymbol{w}^{\tau_i(t)} - \boldsymbol{w}^t \|_2^2 \right)$$

$$\leq 2K^2 \sigma^2 \bar{\eta}^2 + 12K^3 \bar{\eta}^2 \mathbb{E} \| \nabla F(\boldsymbol{w}^t) \|_2^2 + \frac{12L^2 K^3}{n} \bar{\eta}^2 \sum_{i \in I_t} \mathbb{E} \| \boldsymbol{w}^{\tau_i(t)} - \boldsymbol{w}^t \|_2^2,$$

as desired. $\qquad \square$

**Lemma 5.** *Suppose that Assumptions 2, 3, and 4 hold. Then, it holds for all $t \geq 0$ and $i \in I_t$ that*

$$\mathbb{E} \| \boldsymbol{w}^t - \boldsymbol{w}^{\tau_i(t)} \|_2^2 \leq 2K\lambda\eta^2 \bar{\eta}^2 \frac{\sigma^2}{n} + 2\lambda\eta^2 \bar{\eta}^2 \sum_{s=(t-\lambda)_+}^{t-1} \mathbb{E} \left\| \frac{1}{n} \sum_{j \in I_s} \sum_{k=0}^{K-1} \nabla F(\boldsymbol{w}_j^{\tau_j(s),k}) \right\|_2^2.$$

*Proof.* For each $i \in I_t$, we consider the following two cases: i) $\tau_i(t) = t$. Then, we have $\mathbb{E} \| \boldsymbol{w}^t - \boldsymbol{w}^{\tau_i(t)} \|_2^2 = 0$. ii) $\tau_i(t) < t$. In view of (5) and (6), the server's global iteration in round $s \in [\tau_i(t), t-1]$ can be written as

$$\boldsymbol{w}^{s+1} = \boldsymbol{w}^s - \frac{\eta}{|I_s|} \sum_{j \in I_s} \bar{\eta} \sum_{k=0}^{K-1} \nabla f(\boldsymbol{w}_j^{\tau_j(s),k}; \boldsymbol{\xi}_j^{s,k}).$$

Then, we have

$$\mathbb{E} \| \boldsymbol{w}^t - \boldsymbol{w}^{\tau_i(t)} \|_2^2$$

$$= \mathbb{E} \left\| \sum_{s=\tau_i(t)}^{t-1} (\boldsymbol{w}^{s+1} - \boldsymbol{w}^s) \right\|_2^2$$

$$= \mathbb{E} \left\| \sum_{s=\tau_i(t)}^{t-1} \frac{\eta}{n} \sum_{j \in I_s} \sum_{k=0}^{K-1} \bar{\eta} \nabla f(\boldsymbol{w}_j^{\tau_j(s),k}; \boldsymbol{\xi}_j^{s,k}) \right\|_2^2$$

$$= \mathbb{E} \left\| \sum_{s=\tau_i(t)}^{t-1} \frac{\eta}{n} \sum_{j \in I_s} \sum_{k=0}^{K-1} \bar{\eta} \left( \nabla f(\boldsymbol{w}_j^{\tau_j(s),k}; \boldsymbol{\xi}_j^{s,k}) - \nabla F(\boldsymbol{w}_j^{\tau_j(s),k}) \right) + \sum_{s=\tau_i(t)}^{t-1} \frac{\eta}{n} \sum_{j \in I_s} \sum_{k=0}^{K-1} \bar{\eta} \nabla F(\boldsymbol{w}_j^{\tau_j(s),k}) \right\|_2^2.$$

Using Fact 2 with $m = 2$ to upper bound the above gives

$$
\mathbb{E}\|\boldsymbol{w}^t - \boldsymbol{w}^{\tau_i(t)}\|_2^2
$$

$$
\leq 2\mathbb{E}\left\| \sum_{s=\tau_i(t)}^{t-1} \frac{\eta}{n} \sum_{j\in I_s} \sum_{k=0}^{K-1} \bar{\eta}\left(\nabla f(\boldsymbol{w}_j^{\tau_j(s),k}; \boldsymbol{\xi}_j^{s,k}) - \nabla F(\boldsymbol{w}_j^{\tau_j(s),k})\right)\right\|_2^2
$$

$$
+ 2\mathbb{E}\left\| \sum_{s=\tau_i(t)}^{t-1} \frac{\eta}{n} \sum_{j\in I_s} \sum_{k=0}^{K-1} \bar{\eta}\nabla F(\boldsymbol{w}_j^{\tau_j(s),k})\right\|_2^2 \tag{29}
$$

$$
\overset{(a)}{\leq} \frac{2\eta^2\bar{\eta}^2}{n^2} nK\lambda\sigma^2 + 2\eta^2\bar{\eta}^2|t-\tau_i(t)| \sum_{s=\tau_i(t)}^{t-1} \mathbb{E}\left\|\frac{1}{n}\sum_{j\in I_s}\sum_{k=0}^{K-1}\nabla F(\boldsymbol{w}_j^{\tau_j(s),k})\right\|_2^2
$$

$$
\overset{(b)}{\leq} 2K\lambda\eta^2\bar{\eta}^2\frac{\sigma^2}{n} + 2\lambda\eta^2\bar{\eta}^2 \sum_{s=(t-\lambda)_+}^{t-1} \mathbb{E}\left\|\frac{1}{n}\sum_{j\in I_s}\sum_{k=0}^{K-1}\nabla F(\boldsymbol{w}_j^{\tau_j(s),k})\right\|_2^2,
$$

where $(a)$ is follows from (19) in Lemma 2 and Fact 2 with $m = |t - \tau_i(t)|$ and $(b)$ is implied by Assumption 4. Combining cases i) and ii) completes the proof. $\qquad\square$

### A.1.2 PUTTING INGREDIENTS TOGETHER

*Proof.* Using the descent lemma implied by Assumption 1, we have

$$
\mathbb{E}[F(\boldsymbol{w}^{t+1})]
$$

$$
\leq \mathbb{E}[F(\boldsymbol{w}^t)] + \mathbb{E}\langle\nabla F(\boldsymbol{w}^t), \boldsymbol{w}^{t+1} - \boldsymbol{w}^t\rangle + \frac{L}{2}\mathbb{E}\|\boldsymbol{w}^{t+1} - \boldsymbol{w}^t\|_2^2
$$

$$
= \mathbb{E}[F(\boldsymbol{w}^t)] + \mathbb{E}\langle\nabla F(\boldsymbol{w}^t), -\boldsymbol{g}^t\rangle + \frac{L}{2}\mathbb{E}\|\boldsymbol{g}^t\|_2^2
$$

$$
= \mathbb{E}[F(\boldsymbol{w}^t)] - \eta\mathbb{E}\langle\nabla F(\boldsymbol{w}^t), \boldsymbol{g}^t - K\bar{\eta}\nabla F(\boldsymbol{w}^t) + K\bar{\eta}\nabla F(\boldsymbol{w}^t)\rangle + \frac{L}{2}\mathbb{E}\|\boldsymbol{g}^t\|_2^2
$$

$$
= \mathbb{E}[F(\boldsymbol{w}^t)] - K\eta\bar{\eta}\mathbb{E}\|\nabla F(\boldsymbol{w}^t)\|_2^2 + \eta\mathbb{E}\langle\nabla F(\boldsymbol{w}^t), \bar{\eta}K\nabla F(\boldsymbol{w}^t) - \boldsymbol{g}^t\rangle + \frac{L}{2}\mathbb{E}\|\boldsymbol{g}^t\|_2^2. \tag{30}
$$

Then, substituting the last two terms using Lemmas 1 and 3, and rearranging give

$$
\mathbb{E}[F(\boldsymbol{w}^{t+1})] - \mathbb{E}[F(\boldsymbol{w}^t)]
$$

$$
\leq - K\eta\bar{\eta}\mathbb{E}\|\nabla F(\boldsymbol{w}^t)\|_2^2 + \frac{K}{2}\eta\bar{\eta}\mathbb{E}\|\nabla F(\boldsymbol{w}^t)\|_2^2
$$

$$
+ \frac{L^2}{n}\eta\bar{\eta} \sum_{i\in I_t}\sum_{k=0}^{K-1}\mathbb{E}\|\boldsymbol{w}_i^{\tau_i(t),k} - \boldsymbol{w}^{\tau_i(t)}\|_2^2 + \frac{L^2 K}{n}\eta\bar{\eta} \sum_{i\in I_t}\mathbb{E}\|\boldsymbol{w}^{\tau_i(t)} - \boldsymbol{w}^t\|_2^2
$$

$$
- \frac{1}{2K}\eta\bar{\eta}\mathbb{E}\left\|\frac{1}{n}\sum_{i\in I_t}\sum_{k=0}^{K-1}\nabla F(\boldsymbol{w}_i^{\tau_i(t),k})\right\|_2^2 + LK\eta^2\bar{\eta}^2\frac{\sigma^2}{n} + L\eta^2\bar{\eta}^2\mathbb{E}\left\|\frac{1}{n}\sum_{i\in I_t}\sum_{k=0}^{K-1}\nabla F(\boldsymbol{w}_i^{\tau_i(t),k})\right\|_2^2
$$

$$
= - \frac{K}{2}\eta\bar{\eta}\mathbb{E}\|\nabla F(\boldsymbol{w}^t)\|_2^2 + LK\eta^2\bar{\eta}^2\frac{\sigma^2}{n}
$$

$$
+ \frac{L^2}{n}\eta\bar{\eta} \sum_{i\in I_t}\sum_{k=0}^{K-1}\mathbb{E}\|\boldsymbol{w}_i^{\tau_i(t),k} - \boldsymbol{w}^{\tau_i(t)}\|_2^2 + \frac{L^2 K}{n}\eta\bar{\eta} \sum_{i\in I_t}\mathbb{E}\|\boldsymbol{w}^{\tau_i(t)} - \boldsymbol{w}^t\|_2^2
$$

$$
- \left(\frac{1}{2K}\eta\bar{\eta} - L\eta^2\bar{\eta}^2\right)\mathbb{E}\left\|\frac{1}{n}\sum_{i\in I_t}\sum_{k=0}^{K-1}\nabla F(\boldsymbol{w}_i^{\tau_i(t),k})\right\|_2^2. \tag{31}
$$

It follows that

$$
\mathbb{E}[F(\boldsymbol{w}^{t+1})] - \mathbb{E}[F(\boldsymbol{w}^t)]
$$

$$
\overset{(a)}{\leq} - \left(\frac{K}{2}\eta\bar{\eta} - 12L^2K^3\eta\bar{\eta}^3\right)\mathbb{E}\|\nabla F(\boldsymbol{w}^t)\|_2^2 + LK\eta^2\bar{\eta}^2\frac{\sigma^2}{n} + 2L^2K^2\eta\bar{\eta}^3\sigma^2
$$

$$
+ \left(\frac{L^2K}{n}\eta\bar{\eta} + \frac{12L^4K^3}{n}\eta\bar{\eta}^3\right)\sum_{i\in I_t}\mathbb{E}\|\boldsymbol{w}^{\tau_i(t)} - \boldsymbol{w}^t\|_2^2
$$

$$
- \left(\frac{1}{2K}\eta\bar{\eta} - L\eta^2\bar{\eta}^2\right)\mathbb{E}\left\|\frac{1}{n}\sum_{i\in I_t}\sum_{k=0}^{K-1}\nabla F(\boldsymbol{w}_i^{\tau_i(t),k})\right\|_2^2.
$$

$$
\overset{(b)}{\leq} - \left(\frac{K}{2}\eta\bar{\eta} - 12L^2K^3\eta\bar{\eta}^3\right)\mathbb{E}\|\nabla F(\boldsymbol{w}^t)\|_2^2 + \left(LK\eta^2\bar{\eta}^2 + 2L^2K^2\lambda\eta^3\bar{\eta}^3 + 24L^4K^4\lambda\eta^3\bar{\eta}^5\right)\frac{\sigma^2}{n}
$$

$$
+ 2L^2K^2\eta\bar{\eta}^3\sigma^2 - \left(\frac{1}{2K}\eta\bar{\eta} - L\eta^2\bar{\eta}^2\right)\mathbb{E}\left\|\frac{1}{n}\sum_{i\in I_t}\sum_{k=0}^{K-1}\nabla F(\boldsymbol{w}_i^{\tau_i(t),k})\right\|_2^2
$$

$$
+ \left(2L^2K\lambda\eta^3\bar{\eta}^3 + 24L^4K^3\lambda\eta^3\bar{\eta}^5\right)\sum_{s=(t-\lambda)_+}^{t-1}\mathbb{E}\left\|\frac{1}{n}\sum_{i\in I_s}\sum_{k=0}^{K-1}\nabla F(\boldsymbol{w}^{\tau_i(s),k})\right\|_2^2
$$

$$
\overset{(c)}{\leq} - \frac{K}{4}\eta\bar{\eta}\mathbb{E}\|\nabla F(\boldsymbol{w}^t)\|_2^2 + 2LK\eta^2\bar{\eta}^2\frac{\sigma^2}{n} + 2L^2K^2\eta\bar{\eta}^3\sigma^2 - \frac{1}{4K}\eta\bar{\eta}\mathbb{E}\left\|\frac{1}{n}\sum_{i\in I_t}\sum_{k=0}^{K-1}\nabla F(\boldsymbol{w}_i^{\tau_i(t),k})\right\|_2^2
$$

$$
+ 4L^2K\lambda\eta^3\bar{\eta}^3\sum_{s=(t-\lambda)_+}^{t-1}\mathbb{E}\left\|\frac{1}{n}\sum_{i\in I_s}\sum_{k=0}^{K-1}\nabla F(\boldsymbol{w}^{\tau_i(s),k})\right\|_2^2, \tag{32}
$$

where $(a)$ uses Lemma 4, $(b)$ uses Lemma 5, $(c)$ holds because condition (10) implies the following:

$$
\bar{\eta} \leq \frac{1}{4\sqrt{3}LK} \iff \frac{K}{2}\eta\bar{\eta} - 12L^2K^3\eta\bar{\eta}^3 \geq \frac{K}{4}\eta\bar{\eta},
$$

$$
\bar{\eta} \leq \frac{1}{2\sqrt{2}LK} \iff 2L^2K^2\lambda\eta^3\bar{\eta}^3 + 24L^4K^4\lambda\eta^3\bar{\eta}^5 \leq 4L^2K^2\lambda\eta^3\bar{\eta}^3,
$$

$$
\eta\bar{\eta} \leq \frac{1}{4LK\lambda} \iff LK\eta^2\bar{\eta}^2 + 4L^2K^2\lambda\eta^3\bar{\eta}^3 \leq 2LK\eta^2\bar{\eta}^2,
$$

$$
\eta\bar{\eta} \leq \frac{1}{2LK} \iff \frac{1}{2K}\eta\bar{\eta} - L\eta^2\bar{\eta}^2 \geq \frac{1}{4K}\eta\bar{\eta}.
$$

Summing inequality (32) for $t = 0, 1, \ldots, T-1$ yields

$$
\mathbb{E}[F(\boldsymbol{w}^T)] - F[\boldsymbol{w}^0]
$$

$$
\leq - \frac{K}{4}\eta\bar{\eta}\sum_{t=0}^{T-1}\mathbb{E}\|\nabla F(\boldsymbol{w}^t)\|_2^2 + 2LKT\eta^2\bar{\eta}^2\frac{\sigma^2}{n} + 2L^2K^2T\eta\bar{\eta}^3\sigma^2
$$

$$
- \frac{1}{4K}\eta\bar{\eta}\sum_{t=0}^{T-1}\mathbb{E}\left\|\frac{1}{n}\sum_{i\in I_t}\sum_{k=0}^{K-1}\nabla F(\boldsymbol{w}_i^{\tau_i(t),k})\right\|_2^2
$$

$$
+ 4L^2K\lambda\eta^3\bar{\eta}^3\sum_{t=0}^{T-1}\sum_{s=(t-\lambda)_+}^{t-1}\underbrace{\mathbb{E}\left\|\frac{1}{n}\sum_{i\in I_s}\sum_{k=0}^{K-1}\nabla F(\boldsymbol{w}^{\tau_i(s),k})\right\|_2^2}_{Y^s}. \tag{33}
$$

Note that

$$
\sum_{t=0}^{T-1}\sum_{s=(t-\lambda)_+}^{t-1} Y^s \leq \sum_{t=0}^{T-2}\lambda Y^t \leq \lambda\sum_{t=0}^{T-1}Y^t. \tag{34}
$$

Substituting this into (33) and rearranging yield

$$
\frac{K}{4}\eta\bar{\eta}\sum_{t=0}^{T-1}\mathbb{E}\|\nabla F(\boldsymbol{w}^t)\|_2^2 \le F[\boldsymbol{w}^0] - \mathbb{E}[F(\boldsymbol{w}^T)] + 2LKT\eta^2\bar{\eta}^2\frac{\sigma^2}{n} + 2L^2K^2T\eta\bar{\eta}^3\sigma^2
$$

$$
- \left(\frac{1}{4K}\eta\bar{\eta} - 4L^2K\lambda^2\eta^3\bar{\eta}^3\right)\sum_{t=0}^{T-1}\mathbb{E}\left\|\frac{1}{n}\sum_{i\in I_t}\sum_{k=0}^{K-1}\nabla F(\boldsymbol{w}_i^{\tau_i(t),k})\right\|_2^2
$$

$$
\le F[\boldsymbol{w}^0] - F^* + 2LKT\eta^2\bar{\eta}^2\frac{\sigma^2}{n} + 2L^2K^2T\eta\bar{\eta}^3\sigma^2,
$$

where the last inequality holds because $F^* \le \mathbb{E}[F(\boldsymbol{w}^T)]$ and condition (10) implies that

$$
\eta\bar{\eta} \le \frac{1}{4LK\lambda} \iff \frac{1}{4K}\eta\bar{\eta} - 4L^2K\lambda^2\eta^3\bar{\eta}^3 \ge 0.
$$

Therefore, we obtain

$$
\frac{1}{T}\sum_{t=0}^{T-1}\mathbb{E}\|\nabla F(\boldsymbol{w}^t)\|_2^2 \le \frac{F(\boldsymbol{w}^0) - F^* + 2LKT\eta^2\bar{\eta}^2\frac{\sigma^2}{n} + 2L^2K^2T\eta\bar{\eta}^3\sigma^2}{(KT/4)\eta\bar{\eta}}
$$

$$
= \frac{4(F(\boldsymbol{w}^0) - F^*)}{KT\eta\bar{\eta}} + 8L\eta\bar{\eta}\frac{\sigma^2}{n} + 8L^2K\bar{\eta}^2\sigma^2
$$

$$
= \sqrt{\frac{128L(F(\boldsymbol{w}^0) - F^*)}{nKT}} + \frac{8L}{\sigma^2 KT},
$$

where the second equality holds since $\eta = \sqrt{(F(\boldsymbol{w}^0) - F^*)nK/2}$ and $\bar{\eta} = 1/(\sqrt{\sigma^2 LT}K)$. This completes the proof of Theorem 1. □

## A.2 Proof of Theorem 2

### A.2.1 Technical Lemmas

**Lemma 6.** *Suppose that Assumptions 1 and 2 hold. Then, it hold for all $t \ge 0$ that*

$$
\mathbb{E}\langle\nabla F(\boldsymbol{w}^t), \bar{\eta}K\nabla F(\boldsymbol{w}^t) - \boldsymbol{h}^t\rangle
$$

$$
\le \frac{K}{2}\bar{\eta}\mathbb{E}\|\nabla F(\boldsymbol{w}^t)\|_2^2 + \frac{L^2}{N}\bar{\eta}\sum_{i=1}^{N}\sum_{k=0}^{K-1}\mathbb{E}\|\boldsymbol{w}_i^{\tau_i(t),k} - \boldsymbol{w}^{\tau_i(t)}\|_2^2 + \frac{L^2K}{N}\bar{\eta}\sum_{i=1}^{N}\mathbb{E}\|\boldsymbol{w}^{\tau_i(t)} - \boldsymbol{w}^t\|_2^2
$$

$$
- \frac{1}{2K}\bar{\eta}\mathbb{E}\left\|\frac{1}{N}\sum_{i=1}^{N}\sum_{k=0}^{K-1}\nabla F_i(\boldsymbol{w}_i^{\tau_i(t),k})\right\|_2^2.
$$

*Proof.* Using the local iteration formula (8) of DLSGD-hetero, we have

$$
\Delta_i^t = \boldsymbol{w}_i^{\tau_i(t),0} - \boldsymbol{w}_i^{\tau_i(t),K} = \sum_{k=0}^{K-1}\bar{\eta}\nabla f_i(\boldsymbol{w}_i^{\tau_i(t),k};\boldsymbol{\xi}_i^{t,k}),
$$

which implies that

$$
\boldsymbol{h}^t = \frac{1}{|I_t|}\sum_{i\in I_t}\Delta_i^t = \frac{\bar{\eta}}{n}\sum_{i\in I_t}\sum_{k=0}^{K-1}\nabla f_i(\boldsymbol{w}_i^{\tau_i(t),k};\boldsymbol{\xi}_i^{t,k}). \tag{35}
$$

In the following, we use $\mathbb{E}_X[\cdot]$ to denote the conditional expectation by taking expectation with respect to $X$ while holding other variables constant. Then, we have

$$
\begin{aligned}
\mathbb{E}_{\mathcal{I}_t}[\boldsymbol{h}^t] &= \mathbb{E}_{\mathcal{I}_t}\left[\frac{1}{n}\sum_{i\in\mathcal{I}_t}\Delta_i^t\right] \\
&= \mathbb{E}_{i_1^t,\ldots,i_n^t}\left[\frac{1}{n}\sum_{m=1}^n \Delta_{i_m^t}^t\right] \\
&\overset{(a)}{=} \frac{1}{n}\left(\mathbb{E}_{i_1^t}[\Delta_{i_1^t}^t] + \cdots + \mathbb{E}_{i_n^t}[\Delta_{i_n^t}^t]\right) \\
&\overset{(b)}{=} \mathbb{E}_{i_1^t}[\Delta_{i_1^t}^t] \\
&\overset{(c)}{=} \sum_{i=1}^N \frac{1}{N}\Delta_i^t \\
&= \frac{1}{N}\sum_{i=1}^N \sum_{k=0}^{K-1} \bar{\eta}\nabla f_i(\boldsymbol{w}_i^{\tau_i(t),k};\boldsymbol{\xi}_i^{t,k}),
\end{aligned}
\tag{36}
$$

where $(a)$ and $(b)$ are because $i_1^t,\ldots,i_n^t$ are independent and identically distributed, respectively, $(c)$ is implies by the uniform distribution of $i_1^t$. We remark that the terms $\sum_{k=0}^{K-1}\bar{\eta}\nabla f_i(\boldsymbol{w}_i^{\tau_i(t),k};\boldsymbol{\xi}_i^{t,k})$ in (36) with $i\notin\mathcal{I}_t$ are *virtual*. They are introduced for mathematical convenience, although the calculations associated with these terms may not occur in practical systems. To proceed, we have

$$
\begin{aligned}
&\mathbb{E}\langle\nabla F(\boldsymbol{w}^t),\bar{\eta}K\nabla F(\boldsymbol{w}^t)-\boldsymbol{h}^t\rangle \\
&= \mathbb{E}\langle\nabla F(\boldsymbol{w}^t),\bar{\eta}K\nabla F(\boldsymbol{w}^t)-\mathbb{E}_{\mathcal{I}_t}[\boldsymbol{h}^t]\rangle \\
&= \bar{\eta}\mathbb{E}\left\langle\nabla F(\boldsymbol{w}^t),K\nabla F(\boldsymbol{w}^t)-\frac{1}{N}\sum_{i=1}^N\sum_{k=0}^{K-1}\nabla f_i(\boldsymbol{w}_i^{\tau_i(t),k};\boldsymbol{\xi}_i^{t,k})\right\rangle,
\end{aligned}
$$

where the first equality is due to Fact 1. The remaining proof can be established using nearly identical arguments as the proof of Lemma 1, which is omitted for brevity. $\square$

**Lemma 7.** *Suppose that Assumptions 2, 3, and 4 hold, and $\mathcal{I}_t = \{j_1^t,\ldots,j_n^t\}$ is a multiset whose elements are independently and uniformly sampled from $[N]$ with replacement in round $t$ ($t\geq 0$) of Algorithm 2. Then, it holds for all $t\geq 0$ that*

$$
\mathbb{E}\left\|\sum_{j\in\mathcal{I}_t}\sum_{k=0}^{K-1}\left(\nabla f_j(\boldsymbol{w}_j^{\tau_j(t),k};\boldsymbol{\xi}_j^{t,k})-\nabla F_j(\boldsymbol{w}_j^{\tau_j(t),k})\right)\right\|_2^2 \leq 2nK\sigma^2,
\tag{37}
$$

*Besides, it holds for all $i\in[n]$ and $t\geq 1$ that*

$$
\mathbb{E}\left\|\sum_{s=\tau_i(t)}^{t-1}\sum_{j\in\mathcal{I}_s}\sum_{k=0}^{K-1}\left(\nabla f_j(\boldsymbol{w}_j^{\tau_j(s),k};\boldsymbol{\xi}_j^{s,k})-\nabla F_j(\boldsymbol{w}_j^{\tau_j(s),k})\right)\right\|_2^2 \leq 2nK\lambda\sigma^2.
\tag{38}
$$

*Proof.* For notational simplicity, we let $\phi_j^t := \sum_{k=0}^{K-1} \left( \nabla f_j(\boldsymbol{w}_j^{\tau_j(t),k}; \boldsymbol{\xi}_j^{t,k}) - \nabla F_j(\boldsymbol{w}_j^{\tau_j(t),k}) \right)$ for $t \geq 1$ and $j \in \mathcal{I}_t$. Then, we have

$$
\mathbb{E}_{\mathcal{I}_t} \left\| \sum_{j \in \mathcal{I}_t} \phi_j^t \right\|_2^2 = \mathbb{E}_{\mathcal{I}_t} \left[ \sum_{i,j \in \mathcal{I}_t} \langle \phi_i^t, \phi_j^t \rangle \right]
$$

$$
= \mathbb{E}_{j_1^t, \dots, j_n^t} \left[ \sum_{u,v \in [n]} \left\langle \phi_{j_u^t}^t, \phi_{j_v^t}^t \right\rangle \right]
$$

$$
\overset{(a)}{=} \sum_{u=1}^{n} \mathbb{E}_{j_u^t} \| \phi_{j_u^t}^t \|_2^2 + \sum_{u,v \in [n], u \neq v} \mathbb{E}_{j_u^t, j_v^t} [ \langle \phi_{j_u^t}^t, \phi_{j_v^t}^t \rangle ]
$$

$$
\overset{(b)}{=} \sum_{u=1}^{n} \sum_{j=1}^{N} \frac{1}{N} \| \phi_j^t \|_2^2 + \sum_{u,v \in [n], u \neq v} \sum_{i,j \in [N]} \frac{1}{N^2} \langle \phi_i^t, \phi_j^t \rangle
$$

$$
= \frac{n}{N} \sum_{j=1}^{N} \| \phi_j^t \|_2^2 + \frac{n(n-1)}{N^2} \sum_{i,j \in [N]} \langle \phi_i^t, \phi_j^t \rangle \tag{39}
$$

$$
= \frac{n}{N} \sum_{j=1}^{N} \| \phi_j^t \|_2^2 + \frac{n(n-1)}{N^2} \sum_{j=1}^{N} \| \phi_j^t \|_2^2 + \frac{n(n-1)}{N^2} \sum_{i,j \in [N], i \neq j} \langle \phi_i^t, \phi_j^t \rangle
$$

$$
= \frac{n(N+n-1)}{N^2} \sum_{j=1}^{N} \| \phi_j^t \|_2^2 + \frac{n(n-1)}{N^2} \sum_{i,j \in [N], i \neq j} \langle \phi_i^t, \phi_j^t \rangle, \tag{40}
$$

where $(a)$ and $(b)$ hold because $j_1, \dots, j_n$ are independent and uniformly distributed over $[N]$, respectively. Following similar arguments for showing (20) in the proof of Lemma 2, we see that

$$
\mathbb{E} \left[ \sum_{\substack{i,j \in [N] \\ i \neq j}} \langle \phi_i^t, \phi_j^t \rangle \right]
$$

$$
= \sum_{\substack{i,j \in [N] \\ i \neq j}} \mathbb{E} \left\langle \sum_{k=0}^{K-1} \nabla f_j(\boldsymbol{w}_i^{\tau_i(t),k}; \boldsymbol{\xi}_i^{t,k}) - \nabla F_j(\boldsymbol{w}_i^{\tau_i(t),k}), \sum_{k=0}^{K-1} \nabla f_j(\boldsymbol{w}_j^{\tau_j(t),k}; \boldsymbol{\xi}_j^{t,k}) - \nabla F_j(\boldsymbol{w}_j^{\tau_j(t),k}) \right\rangle
$$

$$
= 0.
$$

This, together with (40), implies that

$$
\mathbb{E} \left\| \sum_{j \in \mathcal{I}_t} \phi_j^t \right\|_2^2 \overset{(a)}{=} \mathbb{E} \left[ \mathbb{E}_{\mathcal{I}_t} \left\| \sum_{j \in \mathcal{I}_t} \sum_{k=0}^{K-1} \left( \nabla f_j(\boldsymbol{w}_j^{\tau_j(t),k}; \boldsymbol{\xi}_j^{t,k}) - \nabla F_j(\boldsymbol{w}_j^{\tau_j(t),k}) \right) \right\|_2^2 \right]
$$

$$
= \mathbb{E} \left[ \frac{n(N+n-1)}{N^2} \sum_{j=1}^{N} \left\| \sum_{k=0}^{K-1} \left( \nabla f_j(\boldsymbol{w}_j^{\tau_j(t),k}; \boldsymbol{\xi}_j^{t,k}) - \nabla F_j(\boldsymbol{w}_j^{\tau_j(t),k}) \right) \right\|_2^2 \right]
$$

$$
\overset{(b)}{=} \frac{n(N+n-1)}{N^2} \sum_{j=1}^{N} \sum_{k=0}^{K-1} \mathbb{E} \left\| \nabla f_j(\boldsymbol{w}_j^{\tau_j(t),k}; \boldsymbol{\xi}_j^{t,k}) - \nabla F_j(\boldsymbol{w}_j^{\tau_j(t),k}) \right\|_2^2
$$

$$
\overset{(c)}{\leq} \frac{n(N+n-1)}{N^2} N K \sigma^2
$$

$$
\overset{(d)}{\leq} 2nK\sigma^2, \tag{41}
$$

where $(a)$ is due to the tower rule, $(b)$ can be verified following similar arguments for showing (23) in the proof of Lemma 2, $(c)$ uses Assumption 3, and $(d)$ holds because $n \leq N$. This completes the proof of (37).

We further note that for any integers $s, r \in [\tau_i(t), t-1]$ such that $s < r$, it follows from Fact 1 that

$$
\mathbb{E}_{\mathcal{I}_r} \left\langle \sum_{j \in \mathcal{I}_s} \phi_j^s, \sum_{j \in \mathcal{I}_r} \phi_j^r \right\rangle = \mathbb{E}_{j_1^r, \ldots, j_n^r} \left\langle \sum_{j \in \mathcal{I}_s} \phi_j^s, \sum_{u=1}^n \phi_{j_u^r}^r \right\rangle
$$

$$
= \left\langle \sum_{j \in \mathcal{I}_s} \phi_j^s, \sum_{u=1}^n \mathbb{E}_{j_u^r}[\phi_{j_u^r}^r] \right\rangle
$$

$$
= \left\langle \sum_{j \in \mathcal{I}_s} \phi_j^s, \sum_{u=1}^n \sum_{j=1}^N \frac{1}{N} \phi_j^r \right\rangle
$$

$$
= \frac{n}{N} \left\langle \sum_{j \in \mathcal{I}_s} \phi_j^s, \sum_{j=1}^N \phi_j^r \right\rangle. \tag{42}
$$

Taking expectation wrt $\boldsymbol{\xi}^r := \{\boldsymbol{\xi}_j^{r,k} : j \in [N], k = 0, 1, \ldots, K-1\}$ for (42) gives

$$
\mathbb{E}_{\boldsymbol{\xi}^r} \left\langle \sum_{j \in \mathcal{I}_s} \phi_j^s, \sum_{j=1}^N \phi_j^r \right\rangle
$$

$$
= \mathbb{E}_{\boldsymbol{\xi}^r} \left\langle \sum_{j \in \mathcal{I}_s} \sum_{k=0}^{K-1} \left( \nabla f_j(\boldsymbol{w}_j^{\tau_j(s),k}; \boldsymbol{\xi}_j^{s,k}) - \nabla F_j(\boldsymbol{w}_j^{\tau_j(s),k}) \right), \sum_{j=1}^N \sum_{k=0}^{K-1} \left( \nabla f_j(\boldsymbol{w}_j^{\tau_j(r),k}; \boldsymbol{\xi}_j^{r,k}) - \nabla F_j(\boldsymbol{w}_j^{\tau_j(r),k}) \right) \right\rangle
$$

$$
= \left\langle \sum_{j \in \mathcal{I}_s} \sum_{k=0}^{K-1} \left( \nabla f_j(\boldsymbol{w}_j^{\tau_j(s),k}; \boldsymbol{\xi}_j^{s,k}) - \nabla F_j(\boldsymbol{w}_j^{\tau_j(s),k}) \right), \sum_{j=1}^N \sum_{k=0}^{K-1} \mathbb{E}_{\boldsymbol{\xi}_j^{r,k}} \left[ \nabla f_j(\boldsymbol{w}_j^{\tau_j(r),k}; \boldsymbol{\xi}_j^{r,k}) - \nabla F_j(\boldsymbol{w}_j^{\tau_j(r),k}) \right] \right\rangle
$$

$$
= 0, \tag{43}
$$

where the second equality follows from Fact 1 and the last equality uses Assumption 2. Combining (42) and (43) implies that for any integers $s, r \in [\tau_i(t), t-1]$ such that $s < r$, we have

$$
\mathbb{E} \left\langle \sum_{j \in \mathcal{I}_s} \phi_j^s, \sum_{j \in \mathcal{I}_r} \phi_j^r \right\rangle = \mathbb{E} \left[ \mathbb{E}_{\mathcal{I}_r} \left\langle \sum_{j \in \mathcal{I}_s} \phi_j^s, \sum_{j \in \mathcal{I}_r} \phi_j^r \right\rangle \right]
$$

$$
= \frac{n}{N} \mathbb{E} \left[ \left\langle \sum_{j \in \mathcal{I}_s} \phi_j^s, \sum_{j=1}^N \phi_j^r \right\rangle \right]
$$

$$
= \frac{n}{N} \mathbb{E} \left[ \mathbb{E}_{\boldsymbol{\xi}^r} \left\langle \sum_{j \in \mathcal{I}_s} \phi_j^s, \sum_{j=1}^N \phi_j^r \right\rangle \right]
$$

$$
= 0.
$$

It follows that

$$
\mathbb{E} \left\| \sum_{s=\tau_i(t)}^{t-1} \sum_{j \in \mathcal{I}_s} \phi_j^s \right\|_2^2 = \sum_{\tau_i(t) \le s, r \le t-1} \mathbb{E} \left\langle \sum_{j \in \mathcal{I}_s} \phi_j^s, \sum_{j \in \mathcal{I}_r} \phi_j^r \right\rangle
$$

$$
= \sum_{s=\tau_i(t)}^{t-1} \mathbb{E} \left\| \sum_{j \in \mathcal{I}_s} \phi_j^s \right\|_2^2
$$

$$
\le 2nK\lambda\sigma^2, \tag{44}
$$

where the inequality follows from (41) and Assumption 4. This completes the proof of (38). $\qquad \square$

**Lemma 8.** *Suppose that Assumptions 2–5 hold. Then, it hold for all $t \ge 0$ that*

$$
\mathbb{E}\|\boldsymbol{h}^t\|_2^2 \le \frac{4K\sigma^2 + 2G^2K^2}{n} \bar{\eta}^2 + 2\bar{\eta}^2 \mathbb{E} \left\| \frac{1}{N} \sum_{i=1}^N \sum_{k=0}^{K-1} \nabla F_i(\boldsymbol{w}_i^{\tau_i(t),k}) \right\|_2^2.
$$

*Proof.* In view of (35), we have

$$\mathbb{E}\|\boldsymbol{h}^t\|_2^2$$

$$= \mathbb{E}\left\|\frac{1}{n}\sum_{i\in\mathcal{I}_t}\sum_{k=0}^{K-1}\bar{\eta}\nabla f_i(\boldsymbol{w}_i^{\tau_i(t),k};\boldsymbol{\xi}_i^{t,k})\right\|_2^2$$

$$= \mathbb{E}\left\|\frac{1}{n}\sum_{i\in\mathcal{I}_t}\sum_{k=0}^{K-1}\bar{\eta}\left(\nabla f_i(\boldsymbol{w}_i^{\tau_i(t),k};\boldsymbol{\xi}_i^{t,k}) - \nabla F_i(\boldsymbol{w}_i^{\tau_i(t),k}) + \nabla F_i(\boldsymbol{w}_i^{\tau_i(t),k})\right)\right\|_2^2$$

$$\overset{(a)}{\leq} 2\mathbb{E}\left\|\frac{1}{n}\sum_{i\in\mathcal{I}_t}\sum_{k=0}^{K-1}\bar{\eta}\left(\nabla f_i(\boldsymbol{w}_i^{\tau_i(t),k};\boldsymbol{\xi}_i^{t,k}) - \nabla F_i(\boldsymbol{w}_i^{\tau_i(t),k})\right)\right\|_2^2 + 2\mathbb{E}\left\|\frac{1}{n}\sum_{i\in\mathcal{I}_t}\sum_{k=0}^{K-1}\bar{\eta}\nabla F_i(\boldsymbol{w}_i^{\tau_i(t),k})\right\|_2^2$$

$$\overset{(b)}{\leq} \frac{2\bar{\eta}^2}{n^2}2nK\sigma^2 + \frac{2\bar{\eta}^2}{n^2}\mathbb{E}\left\|\sum_{i\in\mathcal{I}_t}\sum_{k=0}^{K-1}\nabla F_i(\boldsymbol{w}_i^{\tau_i(t),k})\right\|_2^2. \tag{45}$$

where the $(a)$ uses Fact 2 with $m = 2$, $(b)$ follows from Lemma 7. Following similar arguments for showing (39), we can obtain

$$\mathbb{E}_{\mathcal{I}_t}\left\|\sum_{i\in\mathcal{I}_t}\sum_{k=0}^{K-1}\nabla F_i(\boldsymbol{w}_i^{\tau_i(t),k})\right\|_2^2$$

$$= \frac{n}{N}\sum_{i=1}^{N}\left\|\sum_{k=0}^{K-1}\nabla F_i(\boldsymbol{w}_i^{\tau_i(t),k})\right\|_2^2 + \frac{n(n-1)}{N^2}\sum_{i,j\in[N]}\left\langle\sum_{k=0}^{K-1}\nabla F_i(\boldsymbol{w}_i^{\tau_i(t),k}), \sum_{k=0}^{K-1}\nabla F_i(\boldsymbol{w}_j^{\tau_j(t),k})\right\rangle$$

$$= \frac{n}{N}\sum_{i=1}^{N}\left\|\sum_{k=0}^{K-1}\nabla F_i(\boldsymbol{w}_i^{\tau_i(t),k})\right\|_2^2 + \frac{n(n-1)}{N^2}\left\|\sum_{i=1}^{N}\sum_{k=0}^{K-1}\nabla F_i(\boldsymbol{w}_i^{\tau_i(t),k})\right\|_2^2. \tag{46}$$

Taking total expectation for both sides of (46) yields

$$\mathbb{E}\left\|\sum_{i\in\mathcal{I}_t}\sum_{k=0}^{K-1}\nabla F_i(\boldsymbol{w}_i^{\tau_i(t),k})\right\|_2^2$$

$$= \mathbb{E}\left[\mathbb{E}_{\mathcal{I}_t}\left\|\sum_{i\in\mathcal{I}_t}\sum_{k=0}^{K-1}\nabla F_i(\boldsymbol{w}_i^{\tau_i(t),k})\right\|_2^2\right]$$

$$= \frac{n}{N}\sum_{i=1}^{N}\mathbb{E}\left\|\sum_{k=0}^{K-1}\nabla F_i(\boldsymbol{w}_i^{\tau_i(t),k})\right\|_2^2 + \frac{n(n-1)}{N^2}\mathbb{E}\left\|\sum_{i=1}^{N}\sum_{k=0}^{K-1}\nabla F_i(\boldsymbol{w}_i^{\tau_i(t),k})\right\|_2^2. \tag{47}$$

Plugging (47) back into (45) gives

$$\mathbb{E}\|\boldsymbol{h}^t\|_2^2$$

$$\leq 4K\bar{\eta}^2\frac{\sigma^2}{n} + \frac{2\bar{\eta}^2}{nN}\sum_{i=1}^{N}\mathbb{E}\left\|\sum_{k=0}^{K-1}\nabla F_i(\boldsymbol{w}_i^{\tau_i(t),k})\right\|_2^2 + \frac{2(n-1)\bar{\eta}^2}{nN^2}\mathbb{E}\left\|\sum_{i=1}^{N}\sum_{k=0}^{K-1}\nabla F_i(\boldsymbol{w}_i^{\tau_i(t),k})\right\|_2^2$$

$$\leq 4K\bar{\eta}^2\frac{\sigma^2}{n} + \frac{2\bar{\eta}^2}{nN}\sum_{i=1}^{N}\sum_{k=0}^{K-1}K\mathbb{E}\|\nabla F_i(\boldsymbol{w}_i^{\tau_i(t),k})\|_2^2 + \frac{2\bar{\eta}^2}{N^2}\mathbb{E}\left\|\sum_{i=1}^{N}\sum_{k=0}^{K-1}\nabla F_i(\boldsymbol{w}_i^{\tau_i(t),k})\right\|_2^2$$

$$\leq 4K\bar{\eta}^2\frac{\sigma^2}{n} + \frac{2G^2K^2\bar{\eta}^2}{n} + 2\bar{\eta}^2\mathbb{E}\left\|\frac{1}{N}\sum_{i=1}^{N}\sum_{k=0}^{K-1}\nabla F_i(\boldsymbol{w}_i^{\tau_i(t),k})\right\|_2^2, \tag{48}$$

where the second inequality follows from Fact 2 with $m = K$ and the third inequality holds due to Assumption 5. This completes the proof. $\qquad\square$

**Lemma 9.** *Suppose that Assumption 1, 2, and 3 hold. Then, it holds for all $t \geq 0$ that*

$$\frac{1}{N} \sum_{i=1}^{N} \sum_{k=0}^{K-1} \mathbb{E} \|\boldsymbol{w}_i^{\tau_i(t),k} - \boldsymbol{w}^{\tau_i(t)}\|_2^2$$

$$\leq (\sigma^2 + 8K\nu^2) 2K^2 \bar{\eta}^2 + 16K^3 \bar{\eta}^2 \mathbb{E} \|\nabla F(\boldsymbol{w}^t)\|_2^2 + \frac{16L^2 K^3}{N} \bar{\eta}^2 \sum_{i=1}^{N} \mathbb{E} \|\boldsymbol{w}^{\tau_i(t)} - \boldsymbol{w}^t\|_2^2.$$

*Proof.* Using the local iteration formula, we have

$$\mathbb{E} \|\boldsymbol{w}_i^{\tau_i(t),k} - \boldsymbol{w}^{\tau_i(t)}\|_2^2$$

$$= \mathbb{E} \|\boldsymbol{w}_i^{\tau_i(t),k-1} - \bar{\eta} \nabla f_i(\boldsymbol{w}_i^{\tau_i(t),k-1}; \boldsymbol{\xi}_i^{t,k-1}) - \boldsymbol{w}^{\tau_i(t)}\|_2^2$$

$$= \mathbb{E} \Big\| \boldsymbol{w}_i^{\tau_i(t),k-1} - \boldsymbol{w}^{\tau_i(t)} - \bar{\eta} \Big( \nabla F_i(\boldsymbol{w}_i^{\tau_i(t),k-1}) - \nabla F_i(\boldsymbol{w}^{\tau_i(t)}) + \nabla F_i(\boldsymbol{w}^{\tau_i(t)}) - \nabla F_i(\boldsymbol{w}^t) + \nabla F_i(\boldsymbol{w}^t)$$

$$- \nabla F(\boldsymbol{w}^t) + \nabla F(\boldsymbol{w}^t) \Big) - \bar{\eta} \Big( \nabla f(\boldsymbol{w}_i^{\tau_i(t),k-1}; \boldsymbol{\xi}_i^{t,k-1}) - \nabla F(\boldsymbol{w}_i^{\tau_i(t),k-1}) \Big) \Big\|_2^2$$

$$= \mathbb{E} \Big\| \boldsymbol{w}_i^{\tau_i(t),k-1} - \boldsymbol{w}^{\tau_i(t)} - \bar{\eta} \Big( \nabla F_i(\boldsymbol{w}_i^{\tau_i(t),k-1}) - \nabla F_i(\boldsymbol{w}^{\tau_i(t)}) + \nabla F_i(\boldsymbol{w}^{\tau_i(t)}) - \nabla F_i(\boldsymbol{w}^t) + \nabla F_i(\boldsymbol{w}^t)$$

$$- \nabla F(\boldsymbol{w}^t) + \nabla F(\boldsymbol{w}^t) \Big) \Big\|_2^2 + \bar{\eta}^2 \mathbb{E} \|\nabla f_i(\boldsymbol{w}_i^{\tau_i(t),k-1}; \boldsymbol{\xi}_i^{t,k-1}) - \nabla F_i(\boldsymbol{w}_i^{\tau_i(t),k-1})\|_2^2,$$

where the last equality holds because Fact 1 and Assumption 2 imply that the cross term is 0, i.e.,

$$2\mathbb{E} \Big\langle \boldsymbol{w}_i^{\tau_i(t),k-1} - \boldsymbol{w}^{\tau_i(t)} - \bar{\eta} \Big( \nabla F_i(\boldsymbol{w}_i^{\tau_i(t),k-1}) - \nabla F_i(\boldsymbol{w}^{\tau_i(t)}) + \nabla F_i(\boldsymbol{w}^{\tau_i(t)}) - \nabla F_i(\boldsymbol{w}^t) + \nabla F_i(\boldsymbol{w}^t)$$

$$- \nabla F(\boldsymbol{w}^t) + \nabla F(\boldsymbol{w}^t) \Big), \nabla f_i(\boldsymbol{w}_i^{\tau_i(t),k-1}; \boldsymbol{\xi}_i^{t,k-1}) - \nabla F_i(\boldsymbol{w}_i^{\tau_i(t),k-1}) \Big\rangle$$

$$= 2\mathbb{E} \Big\langle \boldsymbol{w}_i^{\tau_i(t),k-1} - \boldsymbol{w}^{\tau_i(t)} - \bar{\eta} \Big( \nabla F(\boldsymbol{w}_i^{\tau_i(t),k-1}) - \nabla F_i(\boldsymbol{w}^{\tau_i(t)}) + \nabla F(\boldsymbol{w}^{\tau_i(t)}) - \nabla F_i(\boldsymbol{w}^t) + \nabla F_i(\boldsymbol{w}^t)$$

$$- \nabla F(\boldsymbol{w}^t) + \nabla F(\boldsymbol{w}^t) \Big), \mathbb{E} \Big[ \nabla f_i(\boldsymbol{w}_i^{\tau_i(t),k-1}; \boldsymbol{\xi}_i^{t,k-1}) - \nabla F_i(\boldsymbol{w}_i^{\tau_i(t),k-1}) \Big| \boldsymbol{w}_i^{\tau_i(t),k-1}, \boldsymbol{w}^t \Big] \Big\rangle$$

$$= 0.$$

It follows that

$$\mathbb{E} \|\boldsymbol{w}_i^{\tau_i(t),k} - \boldsymbol{w}^{\tau_i(t)}\|_2^2$$

$$\overset{(a)}{\leq} \bar{\eta}^2 \sigma^2 + \left( 1 + \frac{1}{2K-1} \right) \mathbb{E} \|\boldsymbol{w}_i^{\tau_i(t),k-1} - \boldsymbol{w}^{\tau_i(t)}\|_2^2$$

$$+ 2K\bar{\eta}^2 \mathbb{E} \|\nabla F_i(\boldsymbol{w}_i^{\tau_i(t),k-1}) - \nabla F_i(\boldsymbol{w}^{\tau_i(t)}) + \nabla F_i(\boldsymbol{w}^{\tau_i(t)}) - \nabla F_i(\boldsymbol{w}^t) + \nabla F_i(\boldsymbol{w}^t) - \nabla F(\boldsymbol{w}^t) + \nabla F(\boldsymbol{w}^t)\|_2^2$$

$$\overset{(b)}{\leq} \left( 1 + \frac{1}{2K-1} \right) \mathbb{E} \|\boldsymbol{w}_i^{\tau_i(t),k-1} - \boldsymbol{w}^{\tau_i(t)}\|_2^2 + 8K\bar{\eta}^2 \mathbb{E} \|\nabla F_i(\boldsymbol{w}_i^{\tau_i(t),k-1}) - \nabla F_i(\boldsymbol{w}^{\tau_i(t)})\|_2^2$$

$$+ 8K\bar{\eta}^2 \mathbb{E} \|\nabla F_i(\boldsymbol{w}^{\tau_i(t)}) - \nabla F_i(\boldsymbol{w}^t)\|_2^2 + 8K\bar{\eta}^2 \mathbb{E} \|\nabla F_i(\boldsymbol{w}^t) - \nabla F(\boldsymbol{w}^t)\|_2^2 + 8K\bar{\eta}^2 \mathbb{E} \|\nabla F(\boldsymbol{w}^t)\|_2^2 + \bar{\eta}^2 \sigma^2$$

$$\overset{(c)}{\leq} \left( 1 + \frac{1}{2K-1} + 8L^2 K\bar{\eta}^2 \right) \mathbb{E} \|\boldsymbol{w}_i^{\tau_i(t),k-1} - \boldsymbol{w}^{\tau_i(t)}\|_2^2$$

$$+ 8L^2 K\bar{\eta}^2 \mathbb{E} \|\boldsymbol{w}^{\tau_i(t)} - \boldsymbol{w}^t\|_2^2 + 8K\bar{\eta}^2 \nu^2 + 8K\bar{\eta}^2 \mathbb{E} \|\nabla F(\boldsymbol{w}^t)\|_2^2 + \bar{\eta}^2 \sigma^2$$

$$\overset{(d)}{\leq} \left( 1 + \frac{1}{K-1} \right) \mathbb{E} \|\boldsymbol{w}_i^{\tau_i(t),k-1} - \boldsymbol{w}^{\tau_i(t)}\|_2^2 + (\sigma^2 + 8K\nu^2)\bar{\eta}^2$$

$$+ 8K\bar{\eta}^2 \mathbb{E} \|\nabla F(\boldsymbol{w}^t)\|_2^2 + 8L^2 K\bar{\eta}^2 \mathbb{E} \|\boldsymbol{w}^{\tau_i(t)} - \boldsymbol{w}^t\|_2^2, \tag{49}$$

where $(a)$ uses Assumption 3 and the fact that $\|\boldsymbol{x} + \boldsymbol{y}\|_2^2 \leq (1 + 1/\beta)\|\boldsymbol{x}\|_2^2 + (1 + \beta)\|\boldsymbol{y}\|_2^2$ for vectors $\boldsymbol{x}, \boldsymbol{y}$ and $\beta = 2K - 1$, $(b)$ uses Fact 2 with $n = 3$, $(c)$ is implied by Assumptions 1 and 6, and $(d)$ holds because

$$\bar{\eta} \leq \frac{1}{4LK} \Rightarrow 1 + \frac{1}{2K-1} + 8L^2 K\bar{\eta}^2 \leq 1 + \frac{1}{K-1}.$$

By solving the recursion (49) using a similar approach as (28), we can derive

$$\mathbb{E}\|\boldsymbol{w}_i^{\tau_i(t),k} - \boldsymbol{w}^{\tau_i(t)}\|_2^2$$
$$\leq 2(K-1)\left((\sigma^2 + 8K\nu^2)\bar{\eta}^2 + 8K\bar{\eta}^2\mathbb{E}\|\nabla F(\boldsymbol{w}^t)\|_2^2 + 8L^2K\bar{\eta}^2\mathbb{E}\|\boldsymbol{w}^{\tau_i(t)} - \boldsymbol{w}^t\|_2^2\right). \quad (50)$$

This implies that

$$\frac{1}{N}\sum_{i=1}^{N}\sum_{k=0}^{K-1}\mathbb{E}\|\boldsymbol{w}_i^{\tau_i(t),k} - \boldsymbol{w}^{\tau_i(t)}\|_2^2$$
$$\leq \frac{1}{N}\sum_{i=1}^{N}\sum_{k=0}^{K-1}2(K-1)\left(\sigma^2\bar{\eta}^2 + 8K\bar{\eta}^2\mathbb{E}\|\nabla F(\boldsymbol{w}^t)\|_2^2 + 8L^2K\bar{\eta}^2\mathbb{E}\|\boldsymbol{w}^{\tau_i(t)} - \boldsymbol{w}^t\|_2^2\right)$$
$$\leq (\sigma^2 + 8K\nu^2)2K^2\bar{\eta}^2 + 16K^3\bar{\eta}^2\mathbb{E}\|\nabla F(\boldsymbol{w}^t)\|_2^2 + \frac{16L^2K^3\bar{\eta}^2}{N}\sum_{i=1}^{N}\mathbb{E}\|\boldsymbol{w}^{\tau_i(t)} - \boldsymbol{w}^t\|_2^2,$$

as desired. $\qquad\square$

**Lemma 10.** *Suppose that Assumptions 2, 3, and 4 hold. Then, it holds for all $t \geq 0$ and $i \in \mathcal{I}_t$ that*

$$\mathbb{E}\|\boldsymbol{w}^t - \boldsymbol{w}^{\tau_i(t)}\|_2^2$$
$$\leq \frac{4\sigma^2 K\lambda + 2G^2K^2\lambda^2}{n}\eta^2\bar{\eta}^2 + 2\lambda\eta^2\bar{\eta}^2\sum_{s=(t-\lambda)_+}^{t-1}\mathbb{E}\left\|\frac{1}{N}\sum_{j=1}^{N}\sum_{k=0}^{K-1}\nabla F_j(\boldsymbol{w}_j^{\tau_j(s),k})\right\|_2^2.$$

*Proof.* For each $i \in \mathcal{I}_t$, we consider the following two cases: i) $\tau_i(t) = t$. Then, we have $\mathbb{E}\|\boldsymbol{w}^t - \boldsymbol{w}^{\tau_i(t)}\|_2^2 = 0$. ii) $\tau_i(t) < t$. Then, employing analogous arguments for establishing (29), we can obtain

$$\mathbb{E}\|\boldsymbol{w}^t - \boldsymbol{w}^{\tau_i(t)}\|_2^2$$
$$\leq 2\mathbb{E}\left\|\sum_{s=\tau_i(t)}^{t-1}\frac{\eta}{n}\sum_{j\in\mathcal{I}_s}\sum_{k=0}^{K-1}\bar{\eta}\left(\nabla f_j(\boldsymbol{w}_j^{\tau_j(s),k};\boldsymbol{\xi}_j^{s,k}) - \nabla F_j(\boldsymbol{w}_j^{\tau_j(s),k})\right)\right\|_2^2$$
$$+ 2\mathbb{E}\left\|\sum_{s=\tau_i(t)}^{t-1}\frac{\eta}{n}\sum_{j\in\mathcal{I}_s}\sum_{k=0}^{K-1}\bar{\eta}\nabla F_j(\boldsymbol{w}_j^{\tau_j(s),k})\right\|_2^2$$
$$\overset{(a)}{\leq} \frac{2\eta^2\bar{\eta}^2}{n^2}2nK\lambda\sigma^2 + \frac{2\eta^2\bar{\eta}^2}{n^2}|t-\tau_i(t)|\sum_{s=\tau_i(t)}^{t-1}\mathbb{E}\left\|\sum_{j\in\mathcal{I}_s}\sum_{k=0}^{K-1}\nabla F_j(\boldsymbol{w}_j^{\tau_j(s),k})\right\|_2^2$$
$$\overset{(b)}{\leq} 4K\lambda\eta^2\bar{\eta}^2\frac{\sigma^2}{n} + \frac{2\lambda\eta^2\bar{\eta}^2}{n^2}\sum_{s=(t-\lambda)_+}^{t-1}\mathbb{E}\left\|\frac{1}{n}\sum_{j\in\mathcal{I}_s}\sum_{k=0}^{K-1}\nabla F_j(\boldsymbol{w}_j^{\tau_j(s),k})\right\|_2^2, \quad (51)$$

where $(a)$ employs Lemma 7 and Fact 2 with $m = |t - \tau_i(t)|$ and $(b)$ is implied by Assumption 4. Besides, it has been shown in (47) that

$$\mathbb{E}\left\|\sum_{j\in\mathcal{I}_s}\sum_{k=0}^{K-1}\nabla F_j(\boldsymbol{w}_j^{\tau_j(s),k})\right\|_2^2$$
$$= \frac{n}{N}\sum_{j=1}^{N}\mathbb{E}\left\|\sum_{k=0}^{K-1}\nabla F_i(\boldsymbol{w}_j^{\tau_j(s),k})\right\|_2^2 + \frac{n(n-1)}{N^2}\mathbb{E}\left\|\sum_{j=1}^{N}\sum_{k=0}^{K-1}\nabla F_j(\boldsymbol{w}_j^{\tau_j(s),k})\right\|_2^2.$$

Substituting this back into (51) gives

$$
\mathbb{E}\|\boldsymbol{w}^t - \boldsymbol{w}^{\tau_i(t)}\|_2^2
$$

$$
\leq 4K\lambda\eta^2\bar{\eta}^2\frac{\sigma^2}{n} + \frac{2\lambda}{nN}\eta^2\bar{\eta}^2 \sum_{s=(t-\lambda)_+}^{t-1} \sum_{j=1}^{N} \mathbb{E}\left\|\sum_{k=0}^{K-1}\nabla F_j(\boldsymbol{w}_j^{\tau_j(s),k})\right\|_2^2
$$

$$
+ \frac{2n(n-1)\lambda}{n^2N^2}\eta^2\bar{\eta}^2 \sum_{s=(t-\lambda)_+}^{t-1} \mathbb{E}\left\|\sum_{j=1}^{N}\sum_{k=0}^{K-1}\nabla F_j(\boldsymbol{w}_j^{\tau_j(s),k})\right\|_2^2
$$

$$
\leq 4K\lambda\eta^2\bar{\eta}^2\frac{\sigma^2}{n} + \frac{2\lambda}{nN}\eta^2\bar{\eta}^2 \sum_{s=(t-\lambda)_+}^{t-1} \sum_{j=1}^{N}\sum_{k=0}^{K-1} K\mathbb{E}\|\nabla F_j(\boldsymbol{w}_j^{\tau_j(s),k})\|_2^2
$$

$$
+ \frac{2n(n-1)\lambda}{n^2}\eta^2\bar{\eta}^2 \sum_{s=(t-\lambda)_+}^{t-1} \mathbb{E}\left\|\frac{1}{N}\sum_{j=1}^{N}\sum_{k=0}^{K-1}\nabla F_j(\boldsymbol{w}_j^{\tau_j(s),k})\right\|_2^2
$$

$$
\leq 4K\lambda\eta^2\bar{\eta}^2\frac{\sigma^2}{n} + \frac{2G^2K^2\lambda^2}{n}\eta^2\bar{\eta}^2 + 2\lambda\eta^2\bar{\eta}^2 \sum_{s=(t-\lambda)_+}^{t-1} \mathbb{E}\left\|\frac{1}{N}\sum_{j=1}^{N}\sum_{k=0}^{K-1}\nabla F_j(\boldsymbol{w}_j^{\tau_j(s),k})\right\|_2^2,
$$

where the second inequality follows from Fact 2 with $m = K$ and the third inequality is due to Assumption 5. Combining cases i) and ii) completes the proof. $\qquad\square$

### A.2.2 Putting Ingredients Together

*Proof.* Employing similar approaches as used to establish (30), we have

$$
\mathbb{E}[F(\boldsymbol{w}^{t+1})] \leq \mathbb{E}[F(\boldsymbol{w}^t)] - K\eta\bar{\eta}\mathbb{E}\|\nabla F(\boldsymbol{w}^t)\|_2^2 + \eta\mathbb{E}\langle\nabla F(\boldsymbol{w}^t), \bar{\eta}K\nabla F(\boldsymbol{w}^t) - \boldsymbol{g}^t\rangle + \frac{L}{2}\mathbb{E}\|\boldsymbol{g}^t\|_2^2.
$$

Substituting the last two term in the above inequality by employing Lemmas 6 and 8, we have

$$
\mathbb{E}[F(\boldsymbol{w}^{t+1})] - \mathbb{E}[F(\boldsymbol{w}^t)]
$$

$$
\leq - K\eta\bar{\eta}\mathbb{E}\|\nabla F(\boldsymbol{w}^t)\|_2^2 + \eta\mathbb{E}\langle\nabla F(\boldsymbol{w}^t), \bar{\eta}K\nabla F(\boldsymbol{w}^t) - \boldsymbol{h}^t\rangle + \frac{L}{2}\mathbb{E}\|\boldsymbol{h}^t\|_2^2
$$

$$
\leq - K\eta\bar{\eta}\mathbb{E}\|\nabla F(\boldsymbol{w}^t)\|_2^2 + \frac{K}{2}\eta\bar{\eta}\mathbb{E}\|\nabla F(\boldsymbol{w}^t)\|_2^2 + \frac{L^2}{N}\eta\bar{\eta} \sum_{i=1}^{N}\sum_{k=0}^{K-1} \mathbb{E}\|\boldsymbol{w}_i^{\tau_i(t),k} - \boldsymbol{w}^{\tau_i(t)}\|_2^2
$$

$$
+ \frac{L^2K}{N}\eta\bar{\eta} \sum_{i=1}^{N} \mathbb{E}\|\boldsymbol{w}^{\tau_i(t)} - \boldsymbol{w}^t\|_2^2 - \frac{1}{2K}\eta\bar{\eta}\mathbb{E}\left\|\frac{1}{N}\sum_{i=1}^{N}\sum_{k=0}^{K-1}\nabla F_i(\boldsymbol{w}_i^{\tau_i(t),k})\right\|_2^2
$$

$$
+ \frac{L}{2}\eta^2\left(\frac{4K\sigma^2 + 2G^2K^2}{n}\bar{\eta}^2 + 2\bar{\eta}^2\mathbb{E}\left\|\frac{1}{N}\sum_{i=1}^{N}\sum_{k=0}^{K-1}\nabla F_i(\boldsymbol{w}_i^{\tau_i(t),k})\right\|_2^2\right)
$$

$$
= - \frac{K}{2}\eta\bar{\eta}\mathbb{E}\|\nabla F(\boldsymbol{w}^t)\|_2^2 + \frac{2\sigma^2LK + G^2LK^2}{n}\eta^2\bar{\eta}^2 + L^2\eta\bar{\eta}\frac{1}{N}\sum_{i=1}^{N}\sum_{k=0}^{K-1} \mathbb{E}\|\boldsymbol{w}_i^{\tau_i(t),k} - \boldsymbol{w}^{\tau_i(t)}\|_2^2
$$

$$
+ \frac{L^2K}{N}\eta\bar{\eta} \sum_{i=1}^{N} \mathbb{E}\|\boldsymbol{w}^{\tau_i(t)} - \boldsymbol{w}^t\|_2^2 - \left(\frac{1}{2K}\eta\bar{\eta} - L\eta^2\bar{\eta}^2\right)\mathbb{E}\left\|\frac{1}{N}\sum_{i=1}^{N}\sum_{k=0}^{K-1}\nabla F_i(\boldsymbol{w}_i^{\tau_i(t),k})\right\|_2^2.
$$

Substituting $\frac{1}{N}\sum_{i=1}^{N}\sum_{k=0}^{K-1}\mathbb{E}\|\boldsymbol{w}_i^{\tau_i(t),k} - \boldsymbol{w}^{\tau_i(t)}\|_2^2$ using Lemma 9, we have

$$
\begin{aligned}
&\mathbb{E}[F(\boldsymbol{w}^{t+1})] - \mathbb{E}[F(\boldsymbol{w}^t)]\\
&\leq -\frac{K}{2}\eta\bar{\eta}\mathbb{E}\|\nabla F(\boldsymbol{w}^t)\|_2^2 + \frac{2\sigma^2 LK + G^2 LK^2}{n}\eta^2\bar{\eta}^2\\
&\quad + L^2\eta\bar{\eta}\left((\sigma^2 + 8K\nu^2)2K^2\bar{\eta}^2 + 16K^3\bar{\eta}^2\mathbb{E}\|\nabla F(\boldsymbol{w}^t)\|_2^2 + \frac{16L^2K^3}{N}\bar{\eta}^2\sum_{i=1}^{N}\mathbb{E}\|\boldsymbol{w}^{\tau_i(t)} - \boldsymbol{w}^t\|_2^2\right)\\
&\quad + \frac{L^2K}{N}\eta\bar{\eta}\sum_{i=1}^{N}\mathbb{E}\|\boldsymbol{w}^{\tau_i(t)} - \boldsymbol{w}^t\|_2^2 - \left(\frac{1}{2K}\eta\bar{\eta} - L\eta^2\bar{\eta}^2\right)\mathbb{E}\left\|\frac{1}{N}\sum_{i=1}^{N}\sum_{k=0}^{K-1}\nabla F_i(\boldsymbol{w}_i^{\tau_i(t),k})\right\|_2^2\\
&= -\left(\frac{K}{2}\eta\bar{\eta} - 16L^2K^3\eta\bar{\eta}^3\right)\mathbb{E}\|\nabla F(\boldsymbol{w}^t)\|_2^2 + \frac{2\sigma^2 LK + G^2 LK^2}{n}\eta^2\bar{\eta}^2 + (\sigma^2 + 8K\nu^2)2L^2K^2\eta\bar{\eta}^3\\
&\quad + \left(\frac{L^2K + 16L^4K^3}{N}\eta\bar{\eta} + \frac{16L^4K^3}{N}\eta\bar{\eta}^3\right)\sum_{i=1}^{N}\mathbb{E}\|\boldsymbol{w}^{\tau_i(t)} - \boldsymbol{w}^t\|_2^2\\
&\quad - \left(\frac{1}{2K}\eta\bar{\eta} - L\eta^2\bar{\eta}^2\right)\mathbb{E}\left\|\frac{1}{N}\sum_{i=1}^{N}\sum_{k=0}^{K-1}\nabla F_i(\boldsymbol{w}_i^{\tau_i(t),k})\right\|_2^2\\
&\leq -\frac{K}{4}\eta\bar{\eta}\mathbb{E}\|\nabla F(\boldsymbol{w}^t)\|_2^2 + \frac{2\sigma^2 LK + G^2 LK^2}{n}\eta^2\bar{\eta}^2 + (\sigma^2 + 8K\nu^2)2L^2K^2\eta\bar{\eta}^3\\
&\quad + \frac{2L^2K}{N}\eta\bar{\eta}\sum_{i=1}^{N}\mathbb{E}\|\boldsymbol{w}^{\tau_i(t)} - \boldsymbol{w}^t\|_2^2 - \frac{1}{4K}\eta\bar{\eta}\mathbb{E}\left\|\frac{1}{N}\sum_{i=1}^{N}\sum_{k=0}^{K-1}\nabla F_i(\boldsymbol{w}_i^{\tau_i(t),k})\right\|_2^2,
\end{aligned}
\tag{52}
$$

where the last inequality holds because condition (13) implies the following:

$$
\begin{aligned}
\bar{\eta} &\leq \frac{1}{8LK} \iff \frac{K}{2}\eta\bar{\eta} - 16L^2K^3\eta\bar{\eta}^3 \geq \frac{K}{4}\eta\bar{\eta},\\
\bar{\eta} &\leq \frac{1}{4LK} \iff L^2K\eta\bar{\eta} + 16L^4K^3\eta\bar{\eta}^3 \leq 2L^2K\eta\bar{\eta},\\
\eta\bar{\eta} &\leq \frac{1}{2LK} \iff \frac{1}{2K}\eta\bar{\eta} - L\eta^2\bar{\eta}^2 \geq \frac{1}{4K}\eta\bar{\eta}.
\end{aligned}
$$

Substituting $\frac{1}{N}\sum_{i=1}^{N}\mathbb{E}\|\boldsymbol{w}^{\tau_i(t)} - \boldsymbol{w}^t\|_2^2$ in (52) using Lemma 10, we have

$$
\begin{aligned}
&\mathbb{E}[F(\boldsymbol{w}^{t+1})] - \mathbb{E}[F(\boldsymbol{w}^t)]\\
&\leq -\frac{K}{4}\eta\bar{\eta}\mathbb{E}\|\nabla F(\boldsymbol{w}^t)\|_2^2 + \frac{2\sigma^2 LK + LG^2K^2}{n}\eta^2\bar{\eta}^2 + (\sigma^2 + 8K\nu^2)2L^2K^2\eta\bar{\eta}^3\\
&\quad + \frac{8\sigma^2 L^2K^2\lambda + 4L^2G^2K^3\lambda^2}{n}\eta^3\bar{\eta}^3 + 4L^2K\lambda\eta^3\bar{\eta}^3\sum_{s=(t-\lambda)_+}^{t-1}\mathbb{E}\left\|\frac{1}{N}\sum_{i=1}^{N}\sum_{k=0}^{K-1}\nabla F_i(\boldsymbol{w}_i^{\tau_i(s),k})\right\|_2^2\\
&\quad - \frac{1}{4K}\eta\bar{\eta}\mathbb{E}\left\|\frac{1}{N}\sum_{i=1}^{N}\sum_{k=0}^{K-1}\nabla F_i(\boldsymbol{w}_i^{\tau_i(t),k})\right\|_2^2\\
&\leq -\frac{K}{4}\eta\bar{\eta}\mathbb{E}\|\nabla F(\boldsymbol{w}^t)\|_2^2 + \frac{4\sigma^2 LK + 2LG^2K^2}{n}\eta^2\bar{\eta}^2\\
&\quad + (\sigma^2 + 8K\nu^2)2L^2K^2\eta\bar{\eta}^3 - \frac{1}{4K}\eta\bar{\eta}\mathbb{E}\left\|\frac{1}{N}\sum_{i=1}^{N}\sum_{k=0}^{K-1}\nabla F_i(\boldsymbol{w}_i^{\tau_i(t),k})\right\|_2^2\\
&\quad + 4L^2K\lambda\eta^3\bar{\eta}^3\sum_{s=(t-\lambda)_+}^{t-1}\mathbb{E}\left\|\frac{1}{N}\sum_{i=1}^{N}\sum_{k=0}^{K-1}\nabla F_i(\boldsymbol{w}_i^{\tau_i(s),k})\right\|_2^2,
\end{aligned}
\tag{53}
$$

where the second inequality holds because condition (13) implies that

$$\eta\bar{\eta} \leq \frac{2\sigma^2 + G^2 K}{8\sigma^2 LK\lambda + 4G^2 LK^2\lambda^2}$$

$$\iff \frac{2\sigma^2 LK + LG^2 K^2}{n}\eta^2\bar{\eta}^2 + \frac{8\sigma^2 L^2 K^2\lambda + 4L^2 G^2 K^3\lambda^2}{n}\eta^3\bar{\eta}^3 \leq \frac{4\sigma^2 LK + 2LG^2 K^2}{n}\eta^2\bar{\eta}^2.$$

By summing inequality (53) over $t = 0, 1, \ldots, T-1$, we obtain

$$\mathbb{E}[F(\boldsymbol{w}^T)] - F[\boldsymbol{w}^0]$$

$$\leq -\frac{K}{4}\eta\bar{\eta}\sum_{t=0}^{T-1}\mathbb{E}\|\nabla F(\boldsymbol{w}^t)\|_2^2 + \frac{4\sigma^2 LKT + 2LG^2 K^2 T}{n}\eta^2\bar{\eta}^2 + (\sigma^2 + 8K\nu^2)2L^2 K^2 T\eta\bar{\eta}^3$$

$$- \frac{1}{4K}\eta\bar{\eta}\sum_{t=0}^{T-1}\mathbb{E}\left\|\frac{1}{N}\sum_{i=1}^{N}\sum_{k=0}^{K-1}\nabla F(\boldsymbol{w}_i^{\tau_i(t),k})\right\|_2^2$$

$$+ 4L^2 K\lambda\eta^3\bar{\eta}^3\sum_{t=0}^{T-1}\sum_{s=(t-\lambda)_+}^{t-1}\mathbb{E}\left\|\frac{1}{N}\sum_{i=1}^{N}\sum_{k=0}^{K-1}\nabla F(\boldsymbol{w}^{\tau_i(s),k})\right\|_2^2$$

$$\leq -\frac{K}{4}\eta\bar{\eta}\sum_{t=0}^{T-1}\mathbb{E}\|\nabla F(\boldsymbol{w}^t)\|_2^2 + \frac{4\sigma^2 LKT + 2LG^2 K^2 T}{n}\eta^2\bar{\eta}^2 + (\sigma^2 + 8K\nu^2)2L^2 K^2 T\eta\bar{\eta}^3$$

$$- \left(\frac{1}{4K}\eta\bar{\eta} - 4L^2 K\lambda^2\eta^3\bar{\eta}^3\right)\sum_{t=0}^{T-1}\mathbb{E}\left\|\frac{1}{N}\sum_{i=1}^{N}\sum_{k=0}^{K-1}\nabla F(\boldsymbol{w}_i^{\tau_i(t),k})\right\|_2^2,$$

$$\leq -\frac{K}{4}\eta\bar{\eta}\sum_{t=0}^{T-1}\mathbb{E}\|\nabla F(\boldsymbol{w}^t)\|_2^2 + \frac{4\sigma^2 LKT + 2LG^2 K^2 T}{n}\eta^2\bar{\eta}^2 + (\sigma^2 + 8K\nu^2)2L^2 K^2 T\eta\bar{\eta}^3,$$

where the second inequality uses (34) and the last inequality holds because condition (10) implies that

$$\eta\bar{\eta} \leq \frac{1}{4LK\lambda} \iff \frac{1}{4K}\eta\bar{\eta} - 4L^2 K\lambda^2\eta^3\bar{\eta}^3 \geq 0.$$

Rearranging and using the fact that $F^* \leq \mathbb{E}[F(\boldsymbol{w}^T)]$, we have

$$\frac{1}{T}\sum_{t=0}^{T-1}\mathbb{E}\|\nabla F(\boldsymbol{w}^t)\|_2^2 \leq \frac{F(\boldsymbol{w}^0) - F^* + \frac{4\sigma^2 LKT + 2LG^2 K^2 T}{n}\eta^2\bar{\eta}^2 + (\sigma^2 + 8K\nu^2)2L^2 K^2 T\eta\bar{\eta}^3}{(KT/4)\eta\bar{\eta}}$$

$$= \frac{4(F(\boldsymbol{w}^0) - F^*)}{KT\eta\bar{\eta}} + \frac{16\sigma^2 L + 8LG^2 K}{n}\eta\bar{\eta} + (\sigma^2 + 8K\nu^2)8L^2 K\bar{\eta}^2$$

$$= \sqrt{\frac{256(F(\boldsymbol{w}^0) - F^*)(16\sigma^2 L + 8LG^2 K)}{nKT}} + \frac{(\sigma^2 + 8K\nu^2)8L^2}{(4\sigma^2 L + 2LKG^2)KT},$$

where the second equality follows by substituting $\eta = \sqrt{4nK(F(\boldsymbol{w}^0) - F^*)}$ and $\bar{\eta} = 1/(\sqrt{(4\sigma^2 L + 2LKG^2)T}K)$. This completes the proof of Theorem 2. □

## B SUPPLEMENTARY EXPERIMENT DETAILS AND RESULTS

### B.1 EXPERIMENTAL SETUP

In this subsection, we present comprehensive details of our experiments. Our objective is to assess the effectiveness of our proposed DLSGD-homo and DLSGD-hetero algorithms by comparing them with their respective baselines. We conduct these comparisons in both iid and non-iid scenarios using the FashionMNIST and CIFAR-10 datasets. In the iid setting, each client is equipped with training data containing all classes. This ensures that the data distribution across clients is statistically homogeneous. To introduce statistical heterogeneity in the non-iid setting, we randomly assign two classes of data to each client. This distribution of data among clients helps create a scenario where

the training data across clients differ, thereby increasing the challenge of achieving convergence in the federated learning process.

**Neural Network Architecture.** We adopt a convolutional neural network, which contains two $5 \times 5$ convolutional layers followed by $2 \times 2$ max pooling layers and two fully connected layers with the latent dimensions being $1600$ and $512$, respectively.

**Evaluation Metrics.** The direct evaluation metric for assessing convergence performance is the number of communication rounds required to reach a target accuracy, as demonstrated by our theorems. However, it's important to note that in scenarios where the computation or communication speeds of different clients vary significantly, the actual runtime of a global round can also differ considerably across different algorithms. In light of this, it is more reasonable to evaluate the practical performance using wall-clock time, which encompasses both CPU computation time and communication time. This approach provides a more comprehensive assessment of the overall efficiency of the algorithms in real-world scenarios.

**System Model.** To measure the wall-clock time, we take into account both the computational time for local training at clients and the communication time for downloading the global model and uploading the local updates.

*Computation Process:* Following the approach described in Sun et al. (2023), we assume that the average time for local training can be represented as $T_{\text{comp}} = N_{\text{MAC}}/C_{\text{MAC}}$, where $N_{\text{MAC}}$ denotes the number of floating-point operations required for one-iteration training and $C_{\text{MAC}}$ denotes the computation speed of the fastest client, set to 10GFLOPS. For our experiments, the cost of updating the local models for one iteration using a mini-batch of ten samples is estimated to be 17.0MFLOPs for FashionMNIST and 31.4MFLOPs for CIFAR-10, respectively. Based on these values, we can calculate the computation time required for local training. To simulate the hardware heterogeneity, we assume that the computation speeds of clients follow the uniform distribution over interval $[1, 5]$, where the slowest client is five times slower than the fastest one. This variation in computation speeds allows us to capture the realistic heterogeneity present in real-world distributed and federated learning systems.

*Communication Process:* Furthermore, we make the assumption that clients transmit the global model and local updates using a 5G network with a transmission bandwidth of 400Mbps for both the downlink and uplink. The model sizes for communication between clients and the server are estimated to be 2.2MB for FashionMNIST and 3.53MB for CIFAR-10, respectively. Based on these values, we can compute the communication time between the server and clients. It is worth noting that although we set the same communication time for all clients, the varying computational capabilities of the clients are sufficient to reflect the impact of system heterogeneity. The computational heterogeneity among clients has a significant influence on the overall performance, even when assuming the same communication time for all clients.

**Hyperparameter Selection.** For all the experiments, the value of the global learning rate $\eta$ (respectively, the local learning rate $\bar{\eta}$) is selected from the set $\{0.1, 1.0\}$ (respectively, $\{0.001, 0.005, 0.01, 0.05, 0.1\}$). We report the results using the learning rates that yield the best convergence performance. The best-tuned global and local learning rates in our experiments are provided in Table 1. Note that the AsySG algorithm does not have a local learning rate. In all of the experiments, we utilize a total of $N = 100$ clients to simulate the distributed and federated learning scenarios. Each client performs local iterations with $K = 50$ using the mini-batch SGD with a fixed batch size of 10 in each step. We repeat all the experiments with three different seeds and report the average results.

### B.2 Additional Numerical Results

In addition to Figure 1, we provide numerical results in Figures 3, 4, and 5. The results demonstrate that our proposed DLSGD algorithms are scalable as the number of participating clients $n$ increases. In all the experiments, DLSGD exhibits superiority to the corresponding baselines in terms of training losses and test accuracies.

We also compare the wall-clock time of the considered algorithms to reach the target test accuracies for different numbers of participating clients $n$. As reported in Table 2, DLSGD uses substantially less time than other algorithms in all cases. For instance, in the iid setting with $n = 10$ on CIFAR-

| | Alogrithms | FashionMNIST | | | | | CIFAR-10 | | | | |
|---|---|---|---|---|---|---|---|---|---|---|---|
| | | $n=5$ | $n=10$ | $n=20$ | $n=40$ | $n=80$ | $n=5$ | $n=10$ | $n=20$ | $n=40$ | $n=80$ |
| iid | Local-SGD | N/A | $\eta=1.00$ $\bar\eta=0.10$ | $\eta=1.00$ $\bar\eta=0.01$ | $\eta=1.00$ $\bar\eta=0.01$ | $\eta=1.00$ $\bar\eta=0.01$ | N/A | $\eta=1.00$ $\bar\eta=0.01$ | $\eta=1.00$ $\bar\eta=0.01$ | $\eta=1.00$ $\bar\eta=0.01$ | $\eta=1.00$ $\bar\eta=0.01$ |
| | AsySG | N/A | $\eta=0.10$ | $\eta=0.10$ | $\eta=0.10$ | $\eta=0.10$ | N/A | $\eta=0.10$ | $\eta=0.10$ | $\eta=0.10$ | $\eta=0.10$ |
| | DLSGD-homo | $\eta=0.10$ $\bar\eta=0.05$ | $\eta=0.10$ $\bar\eta=0.05$ | $\eta=0.10$ $\bar\eta=0.05$ | $\eta=0.10$ $\bar\eta=0.05$ | $\eta=1.00$ $\bar\eta=0.10$ | $\eta=0.10$ $\bar\eta=0.05$ | $\eta=1.00$ $\bar\eta=0.05$ | $\eta=0.10$ $\bar\eta=0.01$ | $\eta=0.10$ $\bar\eta=0.05$ | $\eta=1.00$ $\bar\eta=0.01$ |
| non-iid | Generalized FedAvg | N/A | $\eta=0.10$ $\bar\eta=0.05$ | $\eta=0.10$ $\bar\eta=0.05$ | $\eta=1.00$ $\bar\eta=0.01$ | $\eta=0.10$ $\bar\eta=0.10$ | N/A | $\eta=1.00$ $\bar\eta=0.01$ | $\eta=1.00$ $\bar\eta=0.01$ | $\eta=0.10$ $\bar\eta=0.05$ | $\eta=0.10$ $\bar\eta=0.05$ |
| | FedBuff | N/A | $\eta=0.10$ $\bar\eta=0.005$ | $\eta=0.10$ $\bar\eta=0.01$ | $\eta=0.10$ $\bar\eta=0.01$ | $\eta=0.10$ $\bar\eta=0.05$ | N/A | $\eta=0.10$ $\bar\eta=0.01$ | $\eta=0.10$ $\bar\eta=0.01$ | $\eta=0.10$ $\bar\eta=0.01$ | $\eta=0.10$ $\bar\eta=0.05$ |
| | DLSGD-hetero | $\eta=0.10$ $\bar\eta=0.05$ | $\eta=0.10$ $\bar\eta=0.05$ | $\eta=0.10$ $\bar\eta=0.05$ | $\eta=0.10$ $\bar\eta=0.05$ | $\eta=0.10$ $\bar\eta=0.05$ | $\eta=0.10$ $\bar\eta=0.05$ | $\eta=0.10$ $\bar\eta=0.05$ | $\eta=0.10$ $\bar\eta=0.10$ | $\eta=0.10$ $\bar\eta=0.05$ | $\eta=0.10$ $\bar\eta=0.05$ |

Table 1: The best-tuned global learning rate $\eta$ and local learning rate $\bar\eta$ for generating the results in Table 2.

| | Algorithms | Accuracy | FashionMNIST | | | | Accuracy | CIFAR-10 | | | |
|---|---|---|---|---|---|---|---|---|---|---|---|
| | | | $n=10$ | $n=20$ | $n=40$ | $n=80$ | | $n=10$ | $n=20$ | $n=40$ | $n=80$ |
| iid | Local-SGD | 90% | 51.89 | 179.1 | 184.9 | 179.83 | 70% | 302.12 | 411.64 | 282.51 | 287.81 |
| | AsySG | | 335.16 | 297.57 | 231.22 | 215.38 | | 501.60 | 369.08 | 400.92 | 345.79 |
| | DLSGD-homo | | **26.39** | **49.26** | **94.49** | **99.72** | | **54.41** | **108.52** | **207.45** | **154.26** |
| non-iid | Generalized FedAvg | 80% | 437.57 | 447.81 | 211.67 | 443.56 | 60% | 934.11 | 983.46 | 1541.2 | 1495.0 |
| | FedBuff | | 198.54 | 218.26 | 604.93 | 422.88 | | 394.16 | 789.63 | 2008.8 | 1463.5 |
| | DLSGD-hetero | | **103.25** | **144.49** | **196.98** | **236.30** | | **331.60** | **497.51** | **684.31** | **786.85** |

Table 2: The wall-clock time (in seconds) of different algorithms to achieve the target accuracies with $N = 100$ and different values of $n$.

10, the time used for DLSGD-homo to reach 70% accuracy is almost 1/5 of that for Local-SGD, demonstrating the effectiveness of asynchronous updates in DLSGD-homo. In the non-iid settings, FedBuff spends less time than the Generalized FedAvg when $n$ is small, while the advantage diminishes as $n$ increases.

We also conduct performance testing of DLSGD with varying values of local steps $K$. Specifically, we select $K = 20, 50, 100$, and the convergence results over communication rounds are presented in Figure 6. As the value of $K$ increases, we observe that DLSGD-homo demonstrates faster convergence rates, thereby supporting the discussion in Sections 3.1 regarding the impact of local steps on convergence. Likewise, DLSGD-hetero displays a similar pattern when $K$ increases. This behavior can potentially be attributed to the non-dominant term in the convergence bound (14), which has an order of $\mathcal{O}(1/KT)$ and is inversely proportional to $K$. Moreover, we notice a decrease in the test accuracies on the CIFAR-10 dataset when $K = 100$. This observation suggests that increasing the number of local steps may not always result in improved generalization performance.

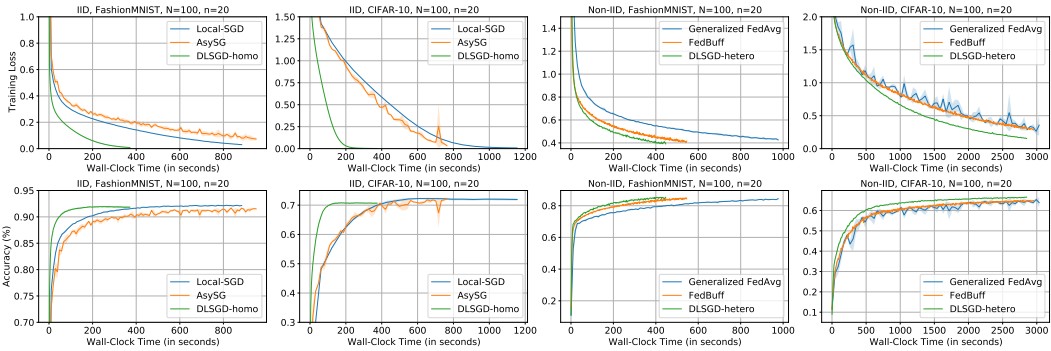

Figure 3: Convergence over wall-clock time of DLSGD and other algorithms with $n = 20$.

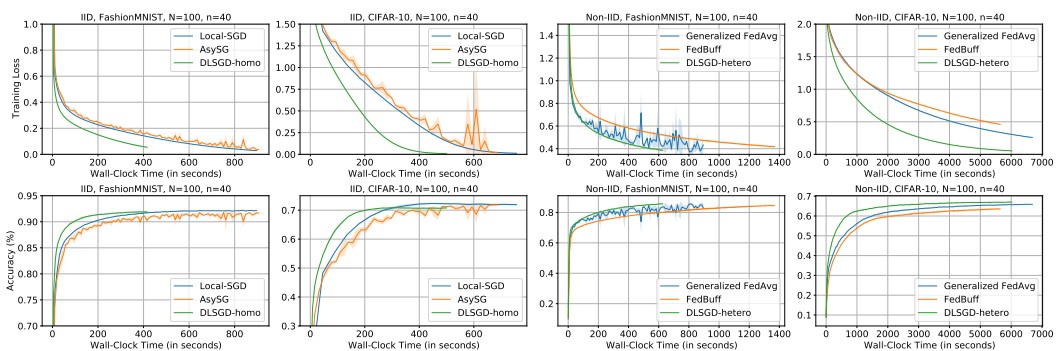

Figure 4: Convergence performance of different algorithms ($N = 100$ and $n = 40$).

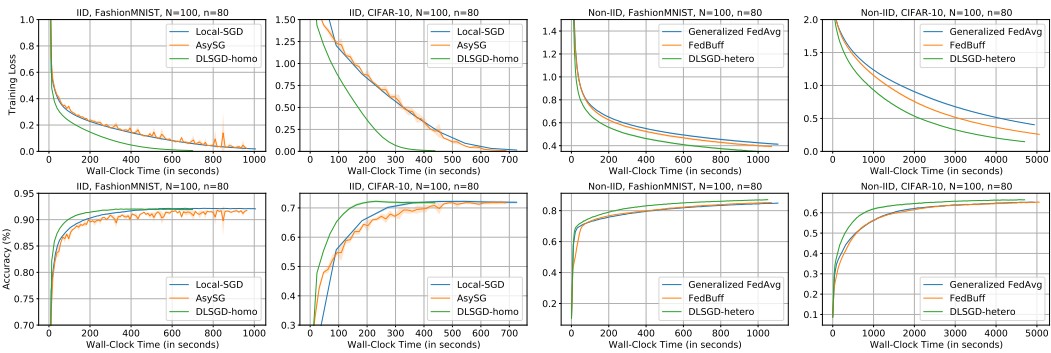

Figure 5: Convergence performance of different algorithms ($N = 100$ and $n = 80$).

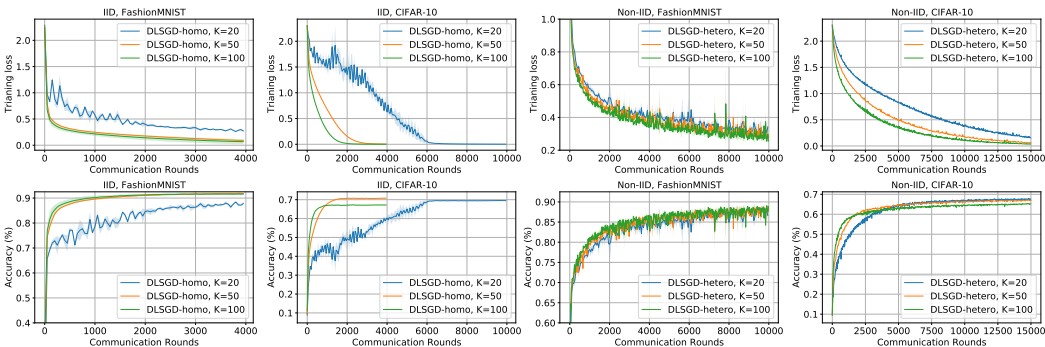

Figure 6: Convergence performance of DLSGD with different values of local steps $K$ ($N = 100$ and $n = 10$).

