# OpenReview forum: "Delayed Local-SGD for Distributed Learning with Linear Speedup"
_ICLR.cc/2024/Conference — Submitted to ICLR 2024_

### Official Review · Reviewer_iNrr · 2023-10-26

**Soundness:** 3 good
**Presentation:** 2 fair
**Contribution:** 2 fair
**Rating:** 5
**Confidence:** 3

**Summary:**

This study examines the FedAvg (local-sgd with partial participation) in the context of communication delays (asynchronous communication). It establishes the method's convergence, and the provided bounds demonstrate an asymptotic linear speedup, where the error decreases linearly with the number of clients.

**Strengths:**

Establishing convergence rate bounds for the delayed local-SGD that match the performance of the non-delayed version asymptotically and imply linear speedup.

**Weaknesses:**

The current form of the algorithm appears inefficient for the heterogeneous case. The server sends its estimate to all nodes, regardless of whether they are sampled or not, potentially wasting communication. Moreover, all clients continuously execute local updates. If a client isn't selected for several rounds, it simply overwrites its old estimate with a newer one derived from a more recent client estimate. This approach seems wasteful and is a significant weakness unless I've misunderstood.

Assuming a bounded gradient is a stringent condition.

**Questions:**

The rate for local-SGD in the heterogeneous case appears to differ from the loca-SGD rate established in the Scaffold paper. Is this rate in terms of communication or computation?

In standard local-SGD implementations, the server sends updates only to clients participating in local updates, leaving others idle. However, in this method, it seems all clients undergo local updates, whether or not they're selected by the server. Additionally, the server dispatches its updated model to every node. One of FedAvg's defining features is its selective communication with clients, which, in practice, cuts communication costs. It might be more practical for the server to send the updated model only to a subset of clients that weren't chosen in the preceding iterations. This way, nodes won't receive a new model until they've completed their local updates based on the last received version. Otherwise, it seems like processing power is expended needlessly.


Moreover, in delayed SGD, once an active client concludes its update, it sends its estimate immediately without waiting for other active clients to finish. Given this, a comprehensive comparison between these models is necessary.

Why is there a distinction between homogeneous and heterogeneous cases? Shouldn't the results from the heterogeneous case inherently encompass those from the homogeneous case?

In theorem 1, it's unclear why the noise variance is in the denominator. For a deterministic scenario where $\sigma=0$ what would the result be? This finding seems counterintuitive.

---

> ### Author Response · Authors · 2023-11-16
>
> **Potential wasting of communication.**
> We require the server to broadcast the global model because all the clients use it for local training, regardless of whether they are sampled or not. In typical scenarios of federated learning, the central server (e.g., a base station) has sufficient power supply and downlink bandwidth to broadcast to all the clients (e.g., smartphones, autonomous vehicles). On the clients' side, the power supply and uplink bandwidth are typically limited, while fortunately only unicast communication to the server is required.
>
> **A client can overwrites its old estimate if not being selected. Clients undergo local updates whether or not they are selected by the server.**
> We can allow clients to continuously execute local updates in practical implementations, so that the potential delay can be minimal, thus reducing the overall runtime. Using this strategy, a client may overwrites its send buffer if it is not selected for several rounds. However, this strategy/implementation is not necessary. Under the circumstances where the clients are energy-limited, we can let the send buffer be written only when the its data are fetched by the server. Both strategies/implementations enjoy the same theoretical property, while may differ in practical runtime efficiency.
>
> **Bounded gradient.**
> Thanks for you question and please kindly refer to our response to the 5th question raised by Reviewer 3asx.
>
> **The local-SGD rate in the heterogeneous case v.s. the local-SGD rate in the Scaffold paper.**
> The reviewer may refer to the (non-delayed) Local-SGD rate in the heterogeneous setting. Indeed, Theorem 2 implies that the Local-SGD has a convergence rate of $\mathcal{O}\left(\frac{\sqrt{K}}{\sqrt{nT}} + \frac{1}{T}\right)$, which is better than the Local-SGD rate of $\mathcal{O}\left(\frac{M}{\sqrt{nKT}} + \frac{1}{T^{2/3}}\right)$ established in the SCAFFOLD paper; cf. Karimireddy et al. (2020, Theorem 1). The rates given by Theorem 1 & 2 are both in terms of the number of global communication rounds $T$.
> *Karimireddy et al., Scaffold: Stochastic controlled averaging for federated learning, ICML 2020.*
>
> **It might be more practical for the server to send the updated model only to a subset of clients that weren't chosen in the preceding iterations.**
> The broadcast is to make sure that all clients have the new global model, so that they can use it for local training once needed. It is feasible that server only send the new global model to a subset of clients that were not selected in the preceding round, and the theoretical guarantees still hold. However, for a client that was selected in a preceding round, the server needs to resend it the latest model and wait for its whole local training period once it is selected again. This incurs longer delay in the local updates and overall system runtime.
>
> **In delayed SGD, once an active client concludes its update, it sends its estimate immediately without waiting for other active clients to finish. The distinction between homogeneous and heterogeneous cases.**
> Our DLSGD-homo for the homogeneous case adopts this participation strategy, i.e., a fast client sends its local update to the server without waiting for other clients. There is no need to worry about the issue of favoring fast clients, because all the clients sample data from the same distribution.
> While in DLSGD-hetero for the heterogeneous case, the clients are selected by the server based on uniform sampling with replacement. According to eq. (36), this is crititcal to ensure that the aggregated gradient is an "unbiased" estimate. Such a scheme resolves the unbalanced treatment of heterogeneous local datasets in FedBuff (Nguyen et al., 2022), and further guarantees convergence with linear speedup.
> Please also kindly refer to our response to Reviewer 3asx's 3rd question for the differences between DLSGD-homo and DLSGD-hetero.
> *Nguyen et al., Federated learning with buffered asynchronous aggregation, AISTATS 2022.*
>
> **In Theorem 1, it's unclear why the noise variance is in the denominator.**
> This is a typo coming from our derivation and we thank the reviewer a lot for pointing out this. The $\sigma^2$ should be eliminated and the bound should be $\mathcal{O}\left(\sqrt{\frac{128(F(w^0)-F^*)}{n K T}} + \frac{8L}{K T}\right)$.
>
> We thank the reviewer for the comments greatly and look forward to any additional feedback you may have.

---

> > ### Comment · Reviewer_iNrr · 2023-11-19
> >
> > I read the reviewers response and I am keeping my original review score unchanged.

---

### Official Review · Reviewer_3asx · 2023-10-30

**Soundness:** 2 fair
**Presentation:** 2 fair
**Contribution:** 2 fair
**Rating:** 3
**Confidence:** 3

**Summary:**

This paper proposes a delayed local-SGD framework for decentralized and federated learning with partial client participation to address the straggler issue. Theoretical results guarantee the linear speedup of the proposed DLSGD under non-convex settings and numerical results validate the efficiency of the proposed algorithm.

**Strengths:**

This paper proposes a delayed local-SGD framework for decentralized and federated learning with partial client participation to address the straggler issue.

**Weaknesses:**

1. The first concern is about the feasibility of the proposed algorithms. To tackle the straggler issue, in each round, the server only needs to wait for the fast n workers to upload the model updates. The question is how the server determines the number n in real implementations. If there is no clear way to determine the number of $n$, the proposed algorithm will lack practical value.

2. How do the receive buffers update in each round $t$, especially for the stragglers? What is the size of it? There seems to be no discussion on it. Please explain it clearly.

3. What is the purpose of send buffer? Can you clearly illustrate the difference between DLSGD-homo and DLSGD-hetero?

4. The reviewer is confused about the linear speedup claimed by this paper. There are total $N$ workers and the server samples $n$ fastest worker per round. The current linear speedup in Theorem 1 and Theorem 2 is for sampler workers $n$, not total number of workers $N$. It is not the linear speedup in previous work cited in this paper. Please explain this.

5. To show the convergence, this paper assumes the bound gradient in Assumption 5. This is actually a weakness compared with most previous work which does not require such an assumption. Please explain why this assumption is necessary.

6. It is weird to this reviewer that the learning rates are set according to $F(w^0)-F^*$ and $\sigma^2$. This is not a common approach.

**Questions:**

see the weakness above.

---

> ### Author Response · Authors · 2023-11-16
>
> 1. The sampling number $n$ needs to be chosen in all other FL algorithms with partial worker participation, such as FedAvg, Scaffold, FedBuff, not just in ours. In our DLSGD, increasing $n$ enhances the system parallelism (leading to less communication rounds) while reducing the client asynchronism (incurring less waiting time). On the contrary, decreasing $n$ reduces the system parallelism while enhances the client asynchronism. Our numerical results would suggest a medium-to-small $n$. However, for all different values of $n$, our DLSGD constantly exhibits better performance than its competitors. Hence, our algorithm is practical in improving the training efficiency.
> 2. In round $t$, the receive buffers are updated after the server broadcasts its global model to all the clients. It is possible that the receive buffer of a certain slow client is overwritten before the model in it is retrieved. In this manner, the receive buffer only needs to store one (latest received) global model, so it has the same size as the model dimension.
> 3. The differences between DLSGD-homo and DLSGD-hetero lie in client sampling and send buffer. In DLSGD-hetero, a client can upload its local updates only when it is sampled by the server. Within a certain round, it is possible that a client has completed its local training while not sampled by the server. In this case, the client just needs to put its local update into the send buffer, and the server can directly fetch it from the send buffer once it selects the client in a subsequent round. This implementation aims to reduce the waiting time of the server, and thus improves the overall runtime efficiency. In DLSGD-homo, there is no mechanism for random client sampling, so the client can directly upload its local update once it completes local training. The difference in their buffering schemes results from the homogeneity and heterogeneity of local datasets.
> 4. For existing distributed/federated learning algorithms with partial client participation, the linear speedup occurs for the (average) number of "participating" clients $n$, rather than the total number of clients $N$. The linear speedup wrt $N$ only occurs in algorithms with "full" client participation. We would kindly refer the reviewer to the discussion and references in Section 2.1 for more details, e.g., Yang et al. (2021), "Achieving linear speedup with partial worker participation in non-iid federated learning". Our DLSGD is a framework with partial client participation, while it subsumes the "full" client participation scheme as a special case by setting $n = N$, which yields the linear speedup wrt $N$.
> 5. For the homogeneous case in Theorem 1, we do not require bounded gradients. For the heterogenous case, we use the bounded gradients to properly control the sum of delayed terms in Lemma 10. Indeed, the bounded gradient assumption remains common in recent works on asynchronous federated learning, such as the following papers, to tackle the challenges introduced by delay. Nevertheless, this is an interesting problem worthy of further study and beyond the accomplishment of this paper.
> *Xie et al., Asynchronous federated optimization, OPT 2019.*
> *Nguyen et al., Federated learning with buffered asynchronous aggregation, AISTATS 2022.*
> *Koloskova et al., Sharper convergence guarantees for asynchronous SGD for distributed and federated learning, NeurIPS 2022.*
>
> 6. The dependence actually does not violate our intuition. On the one hand, the global learning rate is positively correlated with $F(w^0) - F^*$, meaning that the closer initial point to the global optimum, the smaller the global stepsize should be. On the other hand, the local learning rate is negatively correlated with the gradient's variance bound $\sigma^2$, meaning that the noisier the local gradient is, the smaller the local stepsize should taken to avoid "client drift" (as interpreted in the following paper).
> *Karimireddy et al., Scaffold: Stochastic controlled averaging for federated learning, ICML 2020.*
>
> We appreciate the your comments sincerely and anticipate any additional feedback you may have.

---

> > ### Comment · Reviewer_3asx · 2023-11-23
> > **Thank you for your response**
> >
> > Thank you for your response. It is true that most existing federated learning algorithms need to determine how to select clients. However, note that FedAvg does not address stragglers issues. In the synchronous update case without stragglers, FedAvg can randomly sample n clients at each round. However, this will be different in the presence of stragglers. The tradeoff described in the response is also reasonable. However, in order to implement the straggler-resilient FL algorithm, there needs a rule to determine how to select n clients in each round. This will also impact the convergence rate. There are already recent efforts in straggler-resilient FL. Also regarding the bounded gradients, the heterogeneous setting has been already explore, and it will be interesting to explore without such assumption.

---

> > > ### Author Response · Authors · 2023-11-23
> > >
> > > Thank you for your feedback and agreement to our response. Although there have been recent efforts in straggler-resilient / asynchronous FL under the heterogeneous setting, the key distinction of our work is to provably achieve linear speedup that can greatly improve the system efficiency. We would be willing to discuss with the reviewer should there be any other questions.

---

### Official Review · Reviewer_JkMR · 2023-11-01

**Soundness:** 2 fair
**Presentation:** 3 good
**Contribution:** 2 fair
**Rating:** 5
**Confidence:** 4

**Summary:**

This paper proposes a Delayed Local SGD (DLSGD) framework for Local SGD and Federated Learning settings, in the presence of stragglers among clients. The main motivation is that traditional synchronous Local SGD suffers from the straggler effect, where the overall training time is bottlenecked by the slowest clients in each round. This paper aims to address this issue. They propose DLSGD for both homogeneous and heterogeneous cases, where in this algorithm the global aggregation only uses a subset of client updates. This reduces waiting time and improves efficiency. The main difference is that DLSGD-homo uses first-arriving client updates while DLSGD-hetero randomly samples clients. They claim that both DLSGD variants achieve convergence rates comparable to synchronous Local SGD, with linear speedup w.r.t. number of clients. Allows delayed updates without hurting asymptotics. They show the effectiveness of the proposed algorithm on several image classification tasks, compared to local SGD and AsySGD.

**Strengths:**

This paper proposes the novel DLSGD framework for asynchronous distributed and federated learning that reduces waiting times and improves efficiency compared to synchronous Local SGD  and FL methods. A major strength is the rigorous theoretical analysis that proves DLSGD achieves asymptotic convergence rates comparable to Local SGD, despite allowing delayed model updates. Additionally, the paper considers both homogeneous and heterogeneous data settings relevant to parallel optimization and federated learning. The algorithms are intuitive and complemented by proofs that are easy to follow. Comprehensive experiments on image classification tasks validate the practical efficiency gains of DLSGD in terms of a faster decrease in training loss and an increase in test accuracy over wall-clock time compared to basic local SGD and FedAvg baselines. Overall, the combination of an asynchronous framework design, provable guarantees, and empirical gains demonstrates DLSGD's strengths in improving large-scale distributed training both theoretically and practically.

**Weaknesses:**

There are several concerns regarding the proposed algorithm:
1. My main concern is that how the stale gradients can contribute to updating the global model. The gradients from different clients have different starting points of $\boldsymbol{w}^{\tau_i(t),0}$, and hence, averaging them together might have adverse effects and counter-intuitive. This criteria has not been explained in both proposed algorithms.
2. Psudo algorithms seems to be confusing to me. In the heterogenous case, the client only starts updating if it gets a new global model (in the recieve buffer). However, global model only broadcasts after getting all N clients update. Hence, if the client has been selected several times,without getting a new model, how should they update the models and send new gradients to the sever.
3. The heterogenous algorithm is a little bit confusing. When you randomly select the clients, if a slow clients selected multiple times, would that make the whole process slower? If we are willing to wait for those slow clients, why not getting updates (with same computing budget) from all clients that might be faster?
4. DLSGD-homo in section 2.1 is intoduced in a way that each client has access to the "same" local dataset rather than IID access to a shared dataset. Assuming that each client has different seed (otherwise they will perform the same update, which is meaningless), this case is similar to sampling with replacement in IID optimization, which is different from sampling without replacement that normally happens in Local SGD methods. Could you explain how they can be compared?
5. Assumption 3 of bounded variance seems to be too relaxed for this problem. Mostly it is bounded by the norm of the gradients as well.
6. The main claim of the paper is the linear convergence rate with respect to the number of clients $\mathcal{O}\left(\frac{1}{\sqrt{kT}}\right)$, and compare it to local SGD methods' convergence rate. However, there are improved convergence rate for these algorithms in order of $\mathcal{O}\left(\frac{1}{{kT}}\right)$, which has not been discussed nor compared with in this paper. For instance see [A].
7. The experimental results are limited. The datasets selected are good and diverse. However, the methods they compared their proposed algorithms with is not sufficient to make the decision. They should compare with more SOTA federated learning and local SGD methods. Especially FL methods with variance reduction could be useful in this scenario to avoid the fast clients dominate the global model updates.


[A] Haddadpour, Farzin, and Mehrdad Mahdavi. "On the convergence of local descent methods in federated learning." arXiv preprint arXiv:1910.14425 (2019).

**Questions:**

Questions are covered in the last section.

---

> ### Author Response · Authors · 2023-11-16
>
> We thank the reviewer a lot for the detailed comments and questions. Let us respond to them one by one as follows:
>
> 1. Indeed, one of the main technical contributions of our paper is to show that aggregating "stale" gradients evaluated at different points also guarantees convergence. From a macro perspective, the aggregated gradient ${g}^t$ (or ${h}^t$) can be viewed as an inexact approximation of the full gradient $\nabla F({w}^t)$ (up to a scaling factor), and the inexactness/error can be properly bounded (as shown in Lemmas 1 and 6). As a consequence, averaging delayed local updates does not hamper the asymptotic convergence rates compared to their non-delayed counterparts, which require a larger number of iterations to achieve the rates (as demonstrated in Theorems 1 and 2).
> 2. First, the server only needs to receive $n$ clients' update and the new global model is broadcasted to all $N$ clients. Second, note that for the server's procedures in Algorithm 2, the client sampling (line 4) occurs after the new model has been broadcasted to all the clients (line 2 or line 9). Hence, a client always gets a new model even if it is selected several times. We thank the reviewer for raising the confusion, and we will make it clearer in the revised manuscript.
> 3. The clients are sampled uniformly with replacement in the heterogeneous case, so it is possible that a client is selected multiple times. In the realistic settings where $n$ is small compared with $N$, this rarely happens. Therefore, our DLSGD-hetero is always more efficient than its synchronous counterpart FedAvg (Yang et al. 2021) for different choices of $n$; see Figures 1 & 3-5. The reviewer's strategy that always gets updates from faster clients impedes convergence since the heterogeneous local datasets are treated unequally (as discussed in Section 3.2). This can be observed from our numerical comparison with FedBuff (which adopts the scheme that favors fast clients) in Figure 1, where the convergence curves of FedBuff exhibit more drastic oscillations and the convergence rates are less sharp than those of DLSGD-hetero.
> 4. In section 2.1, we mention that DLSGD-homo is applicable to batch data in a shared memory, which means that all the clients access to a shared dataset. In this case, the data points are uniformly sampled by each client with replacement. This actually coincides with what the reviewer says (if we understand the reviewer's confusion correctly).
> 5. Bounded variance is a common assumption that has been made in many recent works; see, e.g., "A Field Guide to Federated Optimization", 2019. The reviewer may refer to a slightly more relaxed assumption like $\mathbb{E} [|| \nabla f_i (w; \xi) - \nabla F_i (w)  ||_2^2 \mid \mathcal{F} ] \leq \sigma^2 (1 + || \nabla F (w) ||_2^2 )$, under which the convergence rates of our algorithms still hold. Indeed, the additional gradient norm in the variance bound does not introduce tricky dominant terms, so our analysis can be readily adapted under this assumption.
> 6. Actually, the convergence rate of $\mathcal{O}(1/T)$ in Paper [A] was established for $L$-smooth functions with the extra Polyak-Łojasiewicz (PL) condition. By contrast, our results hold for general $L$-smooth functions, which subsume PL functions as a subclass. In other words, the faster rate in [A] follows from much stricter conditions, which is not an improvement upon our results.
> 7. In addition to the SOTA federated learning and local SGD methods provided in our paper, we will add comparisons with more recent works in the revised version. Meanwhile, the variance reduction methods are orthogonal to our asynchronous approach. For example, Scaffold also uses the uniform sampling to avoid the fast clients from dominating the global model updates. Besides, it might be possible to incorporate variance reduction techniques (e.g., control variates) into our DLSGD.

---

### Official Review · Reviewer_gXZN · 2023-11-01

**Soundness:** 2 fair
**Presentation:** 2 fair
**Contribution:** 2 fair
**Rating:** 3
**Confidence:** 5

**Summary:**

This paper combines asynchronous sgd and local sgd, and then provide convergence analysis. The novelty is incremental and the convergence analysis is straightforward.

**Strengths:**

The problem studied is important.

This paper gave a good literature review.

**Weaknesses:**

1. The convergence rate of (Stich, 2019) provided in this paper is NOT correct.

2. The novelty is incremental. The convergence analysis is relatively standard. The proof is a straightforward combination of federated learning and asynchronous sgd.

3. The number of iterations has a quadratic dependence on $\lambda$. It is suboptimal. [1] can achieve a sharper bound.

[1] Sharper Convergence Guarantees for Asynchronous SGD for Distributed and Federated Learning

**Questions:**

1. The convergence rate of (Stich, 2019) provided in this paper is NOT correct.

2. The novelty is incremental. The convergence analysis is relatively standard. The proof is a straightforward combination of federated learning and asynchronous sgd.

3. The number of iterations has a quadratic dependence on $\lambda$. It is suboptimal. [1] can achieve a sharper bound.

[1] Sharper Convergence Guarantees for Asynchronous SGD for Distributed and Federated Learning

---

> ### Author Response · Authors · 2023-11-16
>
> **Convergence rate in Stich (2019).**
>
> We thank the reviewer for pointing out this typo. The rate should be $\mathcal{O}(1/nT)$ as the analysis is based on strongly convex objectives.
>
> **Novelty.**
>
> We would like to highlight that the key contribution of our paper is to provide an effective approach to guarantee linear speedup in asynchronous FL. Establishing our linear speedup results relies on a meticulous treatment of the staleness in the analysis, which is not a trivial task and is new to the literature. As we have discussed in Sections 1 and 3,  existing works on asynchronous FL, e.g., Nguyen et al. (2022), Koloskova et al. (2022), fail to maintain linear speedup that is originally possessed by their synchronous counterparts. Our results reveal the efficiency of asynchronous FL with both theoretical and empirical strengths.
>
> Nguyen et al., Federated learning with buffered asynchronous aggregation, AISTATS 2022;
> Koloskova et al., Sharper convergence guarantees for asynchronous SGD for distributed and federated learning, NeurIPS 2022.
>
>
> **Differences between [1] and our paper.**
>
> The asynchronous SGD in [1] (Koloskova et al., 2022) is rather different from our DLSGD in algorithmic techniques, and its theoretical results cannot be directly applied to our approach. We have briefly discussed on [1] in Section 3.2 of our paper, while let us further elaborate here:
>
> i) First and foremost, in the asynchronous SGD [1, Algorithms 1 & 2], the global update takes place once an active client sends its local stochastic gradient to the server. As a consequence, the iteration complexity of $\mathcal{O}(1/\epsilon^2)$ in [1, Theorems 6, 8, & 11] does NOT exhibit linear speedup (we ignore the non-dominant terms for ease of exposition). This is because the gradient estimate in its global iteration utilizes only one client's gradient, leading to high variance. By contrast, our DLSGD aggregates delayed local information from ''multiple'' clients in each global iteration to reduce the estimate's variance, improving the iteration complexity to $\mathcal{O}(1/n\epsilon^2)$ with linear speedup. Since communication efficiency is a major bottleneck in distributed/federated learning, the better iteration complexity of DLSGD than that in [1] would imply substantially reduced overall communication time particularly when $n$ increases.
>
> ii) Second, the asynchronous SGD in [1] does not have local updates, where the clients are only responsible for computing local stochastic gradients. By contrast, our DLSGD adopts combined scheme of local and global updates.
>
> iii) Third, even for the special case when $n=1$ and $K=1$, DLSGD does not reduces to the asynchronous SGD in [1]. Specifically, the asynchronous SGD in [1] is based on the quantity called "concurrency" and its theoretical analyses do apply to our DLSGD.
>
> Again, we thank the reviewer for the comments and look forward to your further feedback.

---

### Meta-Review · Area_Chair_gpqT · 2023-12-04

**Metareview:**

This paper proposes a Delayed Local SGD (DLSGD) framework for Local SGD and Federated Learning settings, in the presence of stragglers among clients. One of the main contributions is the proposal of DLSGD-hetero (Algorithm 2) that is applicable in heterogeneous data scenarios and that addresses the straggler issue by subsampling a set of clients uniformly at random.

The convergence of the algorithm is studied under standard assumptions, and it is proven that the method enjoys linear speedup (in the number of participating devices).

The reviewers noted a couple of typos in the initially submitted version, but the revision addressed these.

Some of the of the reviewer main concerns where
- the limited numerical evaluation (with comparison only to one single baseline)
- ambiguity in the description of the method (although the discussion addressed these)
- tightness of the results (or lacking discussion/ablation study).

Overall, the reviewers found that the paper addresses an important problem. However, further numerical comparisons would be welcome, and discussion on the computational overhead vs. other proposed schemes.

**Justification For Why Not Higher Score:**

The reviewers found that the technical challenges have not been sufficiently highlighted to warrant acceptance based just on the theoretical contributions. The numerical comparisons could be improved to make the contribution stronger.

While I think the work is on good track, the presentation could be a bit more focused to highlight the relevance of the contributions a bit more.

**Justification For Why Not Lower Score:**

N/A

---

### Decision · Program_Chairs · 2024-01-16

Reject